# Allele-specific enhancer interaction at the *Peg3* imprinted domain

**Joomyeong Kim**[1]*, **Wesley D. Frey**[1], **Kaustubh Sharma**[1], **Subash Ghimire**[1], **Ryoichi Teruyama**[1], **Lisa Stubbs**[2]

**1** Department of Biological Sciences, Louisiana State University, Baton Rouge, Louisiana, United States of America, **2** Cell and Developmental Biology, Institute for Genomic Biology, University of Illinois, Urbana, Illinois, United States of America

* jkim@lsu.edu

**Data Availability Statement:** All relevant data are within the paper and its Supporting Information files.

**Funding:** This research was supported by the NIH (National Institute of General Medical Sciences;

## Abstract

The parental allele specificity of mammalian imprinted genes has been evolutionarily well conserved, although its functional constraints and associated mechanisms are not fully understood. In the current study, we generated a mouse mutant with switched active alleles driving the switch from paternal-to-maternal expression for *Peg3* and the maternal-to-paternal expression for *Zim1*. The expression levels of *Peg3* and *Zim1*, but not the spatial expression patterns, within the brain showed clear differences between wild type and mutant animals. We identified putative enhancers localized upstream of *Peg3* that displayed allele-biased DNA methylation, and that also participate in allele-biased chromosomal conformations with regional promoters. Most importantly, these data suggest for the first time that long-distance enhancers may contribute to allelic expression within imprinted domains through allele-biased interactions with regional promoters.

## Introduction

In eutherians, a subset of genes is expressed mainly from one parental allele due to an epigenetic mechanism termed genomic imprinting; there are about 100–200 imprinted genes in mammalian genomes [1,2]. The mono-allelic expression and parental-allele specificity have been well preserved for most imprinted genes during eutherian evolution, although the biological impetus for this dosage and allele specificity is not fully understood. The majority of imprinted genes are expressed in embryos, placentas and brain, and genetic studies have further demonstrated roles in controlling fetal growth rates and maternal-caring behaviors [1,2]. The majority of imprinted genes are found only in the eutherian mammals, and it has been conjectured that genomic imprinting may have emerged in the eutherian lineage to cope with viviparity and placentation [3–6]. Imprinted genes tend to be clustered in specific regions of chromosomes, forming imprinted domains, which are regulated through small genomic regions termed ICRs (Imprinting Control Regions). Genetic studies have also demonstrated that mutations within these ICRs usually disrupt the imprinting and transcription of the individual genes [1,2]. It is, however, currently unknown what other types of DNA elements may be involved in the allelic expression of imprinted genes besides the known ICRs.

R01-GM066225 and R01-GM097074 to J.K.). The
funders had no role in study design, data collection
and analysis, decision to publish, or preparation of
the manuscript.

**Competing interests:** The authors have declared
that no competing interests exist.

The *Peg3* imprinted domain is localized in a 500-kb genomic interval in human chromosome 19q13.4/proximal mouse chromosome 7 that is evolutionarily well conserved among mammals [7–10]. This domain contains paternally expressed *Peg3*, *Usp29*, *Zfp264*, *APeg3* and maternally expressed *Zim1*, *Zim2*, *Zim3* (Fig 1) [10]. As seen in other domains, the imprinting of this domain is controlled through an ICR, the Peg3-DMR (Differentially Methylated Region), which encompasses a 4-kb genomic interval containing the bidirectional promoter for *Peg3* and *Usp29* [11]. According to the results from a mutant allele termed KO2, deletion of this ICR results in complete abrogation of the transcription of paternally expressed *Peg3* and *Usp29*, and the concurrent biallelic expression of *Zim1* through reactivation of its paternal allele [12]. The maternal-specific DNA methylation of this ICR is established during oogenesis through unknown transcription-mediated mechanisms involving an oocyte-specific alternative promoter called U1, which is localized 20-kb upstream of the Peg3-DMR [13]. Maternal deletion of this upstream promoter causes complete removal of oocyte-specific DNA methylation on the ICR, resulting in biallelic expression of *Peg3*/*Usp29* [14]. Overall, this series of studies demonstrated that the imprinting of the *Peg3* domain is mediated through the Peg3-DMR and the U1 alternative promoter.

Besides the two regulatory regions, the *Peg3* domain also contains a large number of evolutionarily conserved regions (ECRs) that are localized in the middle 200-kb transcribed region of *Usp29* [15,16]. These ECRs are usually marked with two histone modifications, H3K4me1 (mono-methylation on lysine 4 of histone 3) and H3K27ac (acetylation on lysine 27 of histone 3), suggesting potential enhancer roles for the transcription of *Peg3* domain. One ECR, ECR18, has been shown to physically interact with the promoter of *Peg3*/*Usp29*, and also recently demonstrated to be a shared enhancer between the paternally expressed *Peg3*/*Usp29* and the maternally expressed *Zim1* [17]. However, the functional contribution of the majority of these ECRs to the imprinting of the *Peg3* domain has not been examined. Recently, we have been able to derive a mouse mutant line with the switching of active alleles, which is driving the maternal expression of *Peg3*/*Usp29* and the paternal expression of *Zim1* [18]. Interestingly, initial investigation of this mutant revealed differences in the expression levels and patterns of genes in the *Peg3* domain. In the current study, we further characterized these expression differences with various experimental approaches. The results suggest that the putative enhancers localized upstream of *Peg3* may be responsible for the observed differences. Further, these enhancers are allele-biased, suggesting that they contribute to allele-specific expression within the *Peg3* domain.

## Results

### Generation of the *Peg3*/*Zim1* mutant with switched active alleles

To generate the mutant with switched active alleles, we used the following mouse breeding scheme involving two mutant alleles, U1 and KO2, targeting the U1 alternative promoter and the Peg3-DMR, respectively (Fig 1). Maternal and paternal transmission of U1 and KO2 mutant alleles is predicted to derive the following 4 genotypes. The first group, referred herein as 'WT', inherits the WT allele from both parents, thus maintaining the normal paternal and maternal expression of *Peg3*/*Usp29* and *Zim1*, respectively (Fig 1B). The second group inherits maternal WT and paternal KO2, subsequently no expression of *Peg3*/*Usp29* but biallelic expression of *Zim1*. The third group inherits maternal U1 and paternal WT, thus biallelic expression of *Peg3*/*Usp29* but no expression of *Zim1*. Finally, the fourth group, referred as 'U1/KO2', inherits maternal U1 and paternal KO2, deriving the maternal and paternal expression of *Peg3*/*Usp29* and *Zim1*, respectively, which are opposite to the parental-specific expression of the imprinted genes in the WT group (Fig 1B). With this breeding scheme, 10 female

## A) Mouse *Peg3* imprinted domain

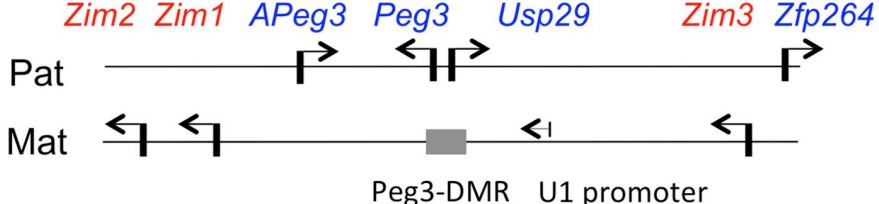

## B) Breeding scheme

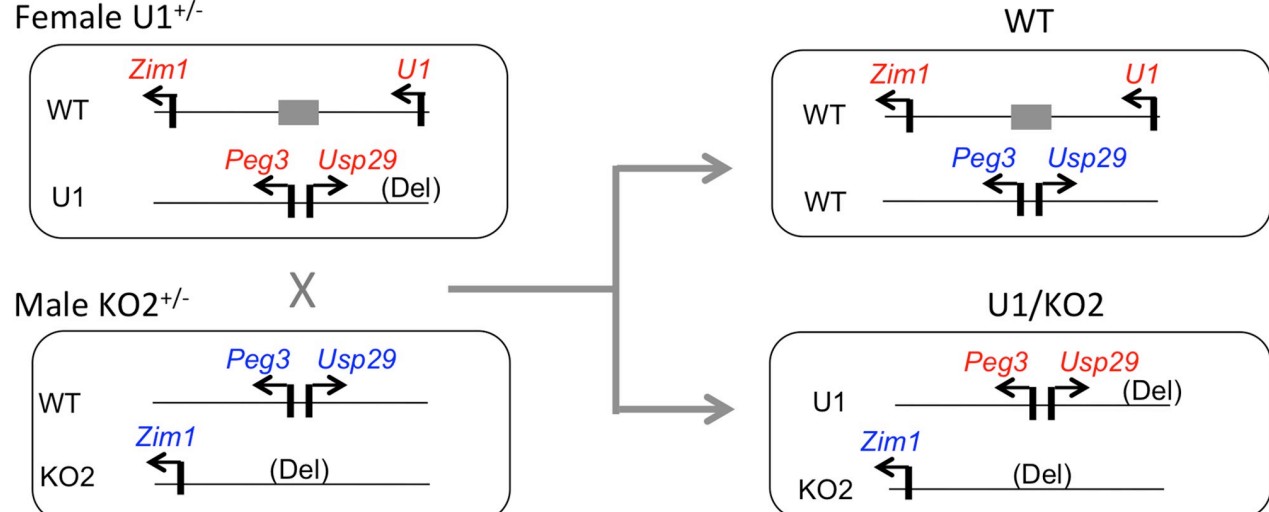

**Fig 1. Generation of the *Peg3/Zim1* mutant with switched active alleles.** (**A**) The genomic structure of the mouse *Peg3* imprinted domain. The paternally and maternally expressed genes are indicated with blue and red, respectively. The direction of each gene is indicated with an arrow. The 4-kb Peg3-DMR is indicated with a grey box, while the oocyte-specific U1 alternative promoter is indicated with a small arrow. (**B**) Breeding scheme. Female U1 heterozygotes were crossed with male KO2 heterozygotes, deriving four possible genotypes. Among these four genotypes, shown are the two genotypes: WT and U1/KO2 with the maternal and paternal expression of *Peg3* and *Zim1*, respectively. The schematic representations for the second and third groups have been omitted for simplicity.

heterozygotes for U1 were crossed with 5 male heterozygotes for KO2, producing a total of 158 pups for 18 litters with the average litter size being 8.72. According to the initial results, these four genotypes were all equally represented among the surviving pups, suggesting no major embryonic lethality associated with any of these four genotypes [18]. Among these four groups, the pups belonging to the two groups, WT and U1/KO2, were further analyzed in the current study: 32 pups (15 females and 17 males) for WT and 36 pups (21 females and 15 males) for U1/KO2 (Fig 1B). More detailed results regarding the breeding experiments are available from the previous study [18].

### Expression level comparison between WT and U1/KO2

The two groups of pups were analyzed in the following manner. We performed a series of qRT-PCR analyses to compare the expression levels of *Peg3* and *Zim1* between WT and U1/KO2 mice (Fig 2). This series of analyses did not include the remaining 5 imprinted genes,

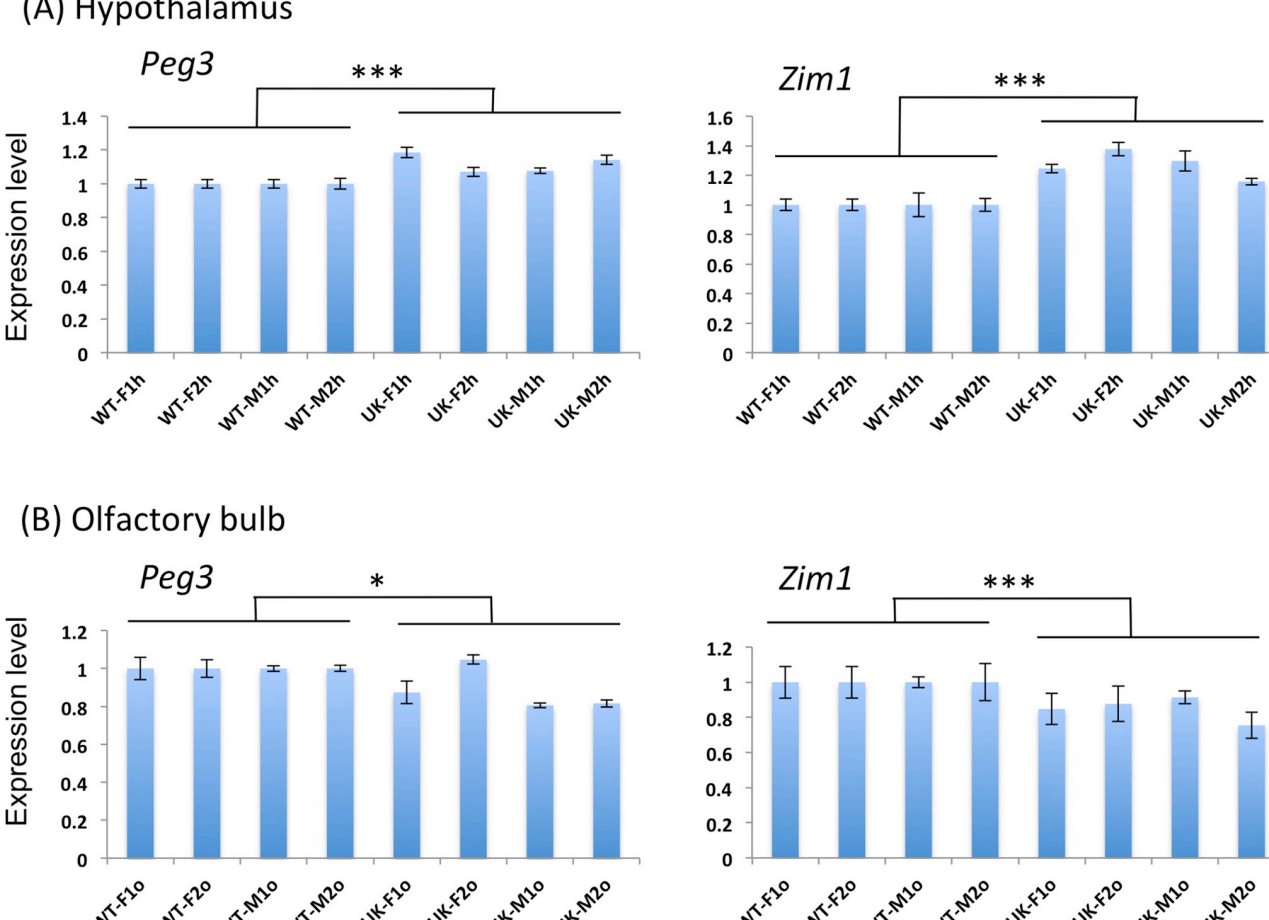

**Fig 2. Expression level comparison of *Peg3* and *Zim1* between WT and U1/KO2.** Expression levels of *Peg3* and *Zim1* were compared between WT and U1/KO2, in short referred to as 'UK.' For this series of analyses, four individual sets of WT and UK were used for isolating total RNA and subsequent cDNA synthesis, which were then used for qRT-PCR analyses. Expression levels of each gene was first normalized with the levels of β-actin, and later compared between WT and UK. The results sets derived from the hypothalamus (**A**) and olfactory bulb (**B**) were presented with the average values and standard deviations. The statistical significance is indicated in the following manner: *, $p < 0.05$; **, $p < 0.01$; and ***, $p < 0.001$.

*Usp29*, *APeg3*, *Zim2*, *Zim3* and *Zfp264*, mainly due to their very low expression levels in the adult tissues that had been selected for the current study. A set of 15 tissues was harvested from each of the 8 adult 2-month-old mice, representing two WT and two U1/KO2 with both sexes (WT-F and WT-M, UK-F and UK-M with UK used as an abbreviation for the U1/KO2 genotype in Fig 2). We isolated total RNA and prepared cDNA from the following tissues: hypothalamus, olfactory bulb, cortex, cerebellum, kidney, and gonadal fat (Fig 2 **and** S1–S3 Files). The results derived from this series of expression analyses are summarized as follows.

First, the expression levels of *Peg3* were 12% higher in the hypothalamus of U1/KO2 than WT (Mann-Whitney U test, *p*-value < 0.00001; Fig 2A). This was also the case for *Zim1*, exhibiting 27% higher expression levels in U1/KO2 than in WT (Mann-Whitney U test, *p*-value < 0.00001). Second, the expression levels of *Peg3* and *Zim1* were lower in two other regions of the brain, including the olfactory bulbs and cerebellum. In the case of olfactory bulbs, the expression levels of *Peg3* and *Zim1* were 12% and 15% lower in U1/KO2 than in WT (Mann-Whitney U test, *p*-value = 0.03 for *Peg3* and *p*-value = 0.0012 for *Zim1*; Fig 2B). This

was also the case for cerebellum with *Peg3* and *Zim1* showing 22% and 18% lower levels in U1/KO2 than in WT (Mann-Whitney U test, *p*-value < 0.00001 for *Peg3* and *p*-value = 0.0455 for *Zim1*; S1 File). On the other hand, the expression levels of *Peg3* and *Zim1* in cortex were indistinguishable mainly due to large variation among individuals with the same genotype and sex (S1 File).

Third, compared to the different brain regions, expression level differences of *Peg3* and *Zim1* between WT and U1/KO2 were neither obvious nor uniform in the other adult tissues, including kidney and gonadal fat (S2 File). In the case of these two tissues, the male set seemed to show some difference between WT and U1/KO2, whereas the female set did not yield any major difference. In the male kidney, the expression levels of *Peg3* and *Zim1* were significantly higher in U1/KO2 than in WT (26% for *Peg3* and 61% for *Zim1*). In the male gonadal fat, on the other hand, the expression levels of *Peg3* and *Zim1* were significantly lower in U1/KO2 than in WT (37% for *Peg3* and 43% for *Zim1*; S2 File). Although the observed differences in these two tissues were more dramatic than those observed from the brain regions, the expression levels of *Peg3* and *Zim1* were 10 to 100-fold lower in these non-neuronal tissues than in the brain regions based on the relative Ct (threshold cycle) values detected through qRT-PCR analyses. Thus, the differences observed in the brain regions might be still more functionally significant than those from the non-neuronal tissues. Fourth, we also analyzed the expression levels of several genes that are expressed and closely linked to the functions of *Peg3*, including *Avp*, *Oxt*, *Oxtr* and *Ghrh* in the hypothalamus set (S3 File). The expression levels of these genes were all higher in U1/KO2 than in WT, although this pattern was detected mainly in the female set. This sex-biased pattern agrees with those observed from the previous study, showing more obvious up-regulation of these genes in the female set [18]. Overall, the expression levels of *Peg3* and *Zim1* displayed clear differences between WT and U1/KO2 in several adult tissues, particularly in the three regions of adult brain, including the hypothalamus, olfactory bulbs and cerebellum.

## Spatial expression pattern of *Zim1* between WT and U1/KO2

The known functions of *Peg3* are closely associated with the physiological roles played by the hypothalamus, including milk provision, neonatal growth, and nurturing behaviors [19–22]. The expression levels of both *Peg3* and *Zim1* in the hypothalamus are also the highest among the different regions of the adult brains. Thus, we previously performed an initial series of immunostaining experiments to monitor the spatial expression patterns of *Peg3* with the sectioned samples prepared from the four genotypes. The data revealed the overall ubiquitous expression of *Peg3* within the adult brains, with no major difference in the spatial expression patterns among the 4 genotypes [18]. To expand on these previous results, we performed a similar series of immunostaining experiments to monitor the spatial expression patterns of *Zim1*, which are known to be more restricted than those of *Peg3* in terms of developmental stage and tissue specificity [23]. This series of immunostaining experiments used a set of 8 adult 2-month-old mice representing each of the four genotypes with both sexes. The harvested brains were sectioned and subsequently immunostained with an anti-ZIM1 polyclonal antibody (Fig 3 **and** S4 File). The results from this series of analyses are as follows. First, the main expression sites of *Zim1* within the hypothalamus include the paraventricular nucleus (PVN), supraoptic nucleus (SON), and the ependymal cells of the 3rd ventricles. The detection in the ependymal cells appeared to be unique to *Zim1*, since similar expression patterns have not been observed from the immunostaining with anti-PEG3 antibody [18,24]. In contrast, the expression patterns within the PVN and SON were very similar to those observed from *Peg3*. This similar pattern between *Peg3* and *Zim1* also supports the prediction that these two genes

## A) WT-Female

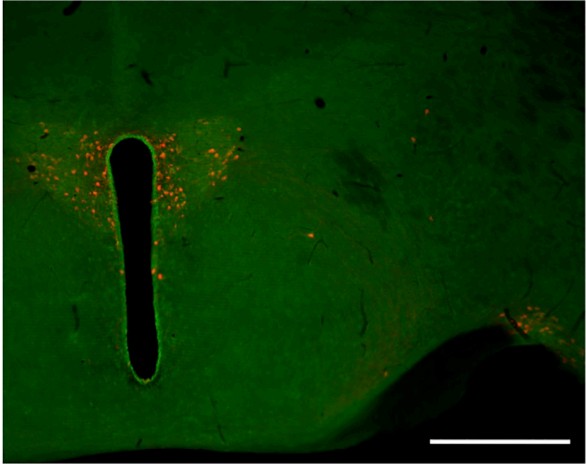

## B) U1/KO2-Female

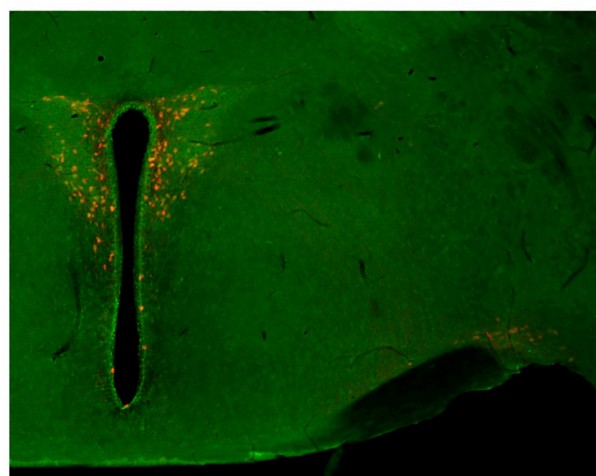

## C) WT-Male

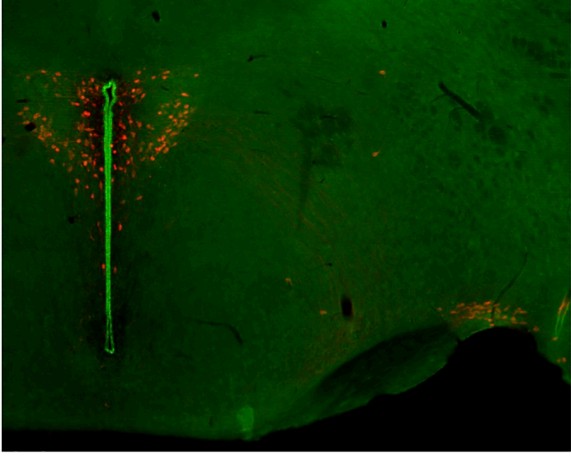

## D) U1/KO2-Male

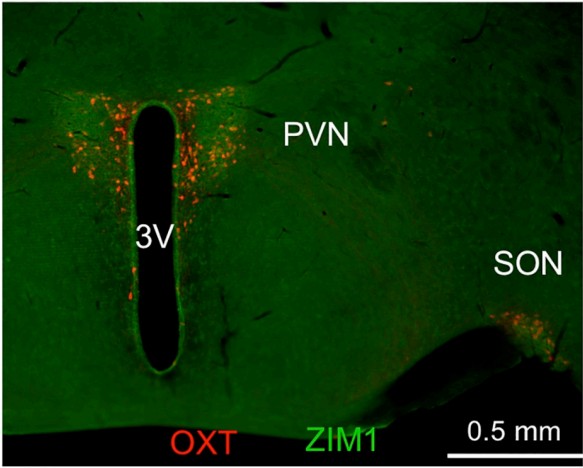

**Fig 3. Spatial expression patterns of *Zim1* within the hypothalamus of WT and U1/KO2.** Immunostaining of ZIM1 (green) and OXT (red) were performed using the hypothalamus prepared from a set of 2-month-old mice of WT and U1/KO2 with two sexes (**A-D**). OXT-immunoreactive neuronal cells are detected mainly within the PVN (Paraventicular nucleus) and SON (Supraoptic nucleus) areas of the hypothalamus. ZIM1-immunoreactive neuron cells are also detected within the similar areas of the hypothalamus with higher levels being detected in PVN, SON and the ependymal cells of the 3rd ventricles. However, there was no major difference between WT and U1/KO2.

may share long-distance enhancers. According to the surveys of the ENCODE dataset and also our own results [16], the middle 200-kb region has the majority of potential enhancers within this imprinted domain. Thus, we believe that the shared enhancers are likely localized within the middle 200-kb region. (Fig 4) [15,16]. Second, the spatial expression patterns of *Zim1* were also similar with no major difference between WT and U1/KO2: both groups displayed the highest expression within PVN, SON and the ependymal cells (Fig 3). Thus, the spatial expression patterns of *Zim1* within the hypothalamus were not affected by switching of the active allele, from maternal to paternal. This further suggests that allele switching does not disturb the transcriptional programs that direct the spatial expression pattern of *Zim1*, at least within the adult hypothalamus.

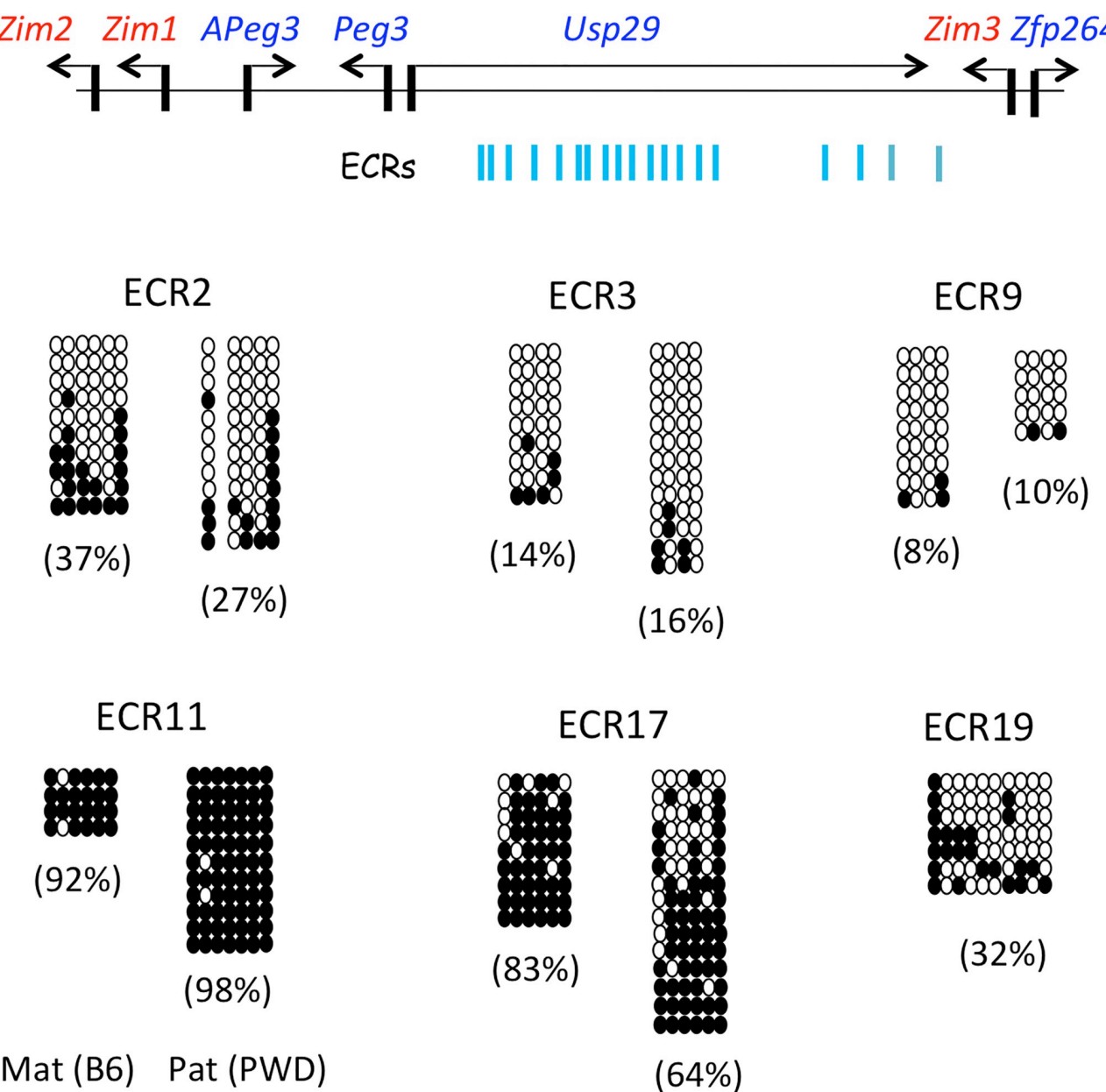

**Fig 4. DNA methylation levels of several ECRs localized upstream of *Peg3*.** The schematic diagram shows the relative positions of 19 ECRs (Evolutionarily Conserved Regions) within the mouse *Peg3* domain (upper panel). The neonatal brain of F1 mice derived from the crossing of C57BL/6J (B6) females and PWD/PhJ (PWD) males were used for isolating genomic DNA, which were then used for the analyses of DNA methylation levels. The DNA methylation levels of six individual ECRs were summarized and presented with open and closed circles indicating unmethylated and methylated CpG sites (lower panel). The methylation levels of each allele for a given ECR were calculated and presented. In the case of ECR19, a presumable SNP was not found from the F1 hybrid that had been used for the current study, thus the results were presented without allele sorting.

### DNA methylation levels of ECRs

To further characterize the expression difference of *Peg3* and *Zim1* between the two genotypes, we tested potential involvement of the 19 ECRs that are localized in the 200-kb upstream region of *Peg3* (Fig 4). As an initial step, we measured the allele-specific DNA methylation levels of the ECRs with the following strategy. In brief, F1 hybrid mice derived from the crossing of male PWD/PhJ and female C57BL/6J were used to differentiate the maternal and paternal

allele using single nucleotide polymorphisms (SNPs) detected between the two strains. Genomic DNA isolated from neonatal brains was first treated with bisulfite conversion protocol [25,26], and subsequently amplified with the 6 individual sets of primers targeting the following ECRs: ECR2, ECR3, ECR9, ECR11, ECR17, and ECR19. The remaining ECRs were not included in the current survey, since they do not have more than 3 CpG sites or SNPs between C57BL/6J and PWD/PhJ. The amplified PCR products were individually cloned and sequenced. The results are summarized as follows. First, ECR2, ECR3, ECR9 and ECR19 displayed relatively low levels of DNA methylation in neonatal brain, ranging from 8 to 37%. In contrast, ECR11 and ECR17 showed high levels of DNA methylation, ranging from 64 to 98%. Second, ECR2 and ECR17 exhibited different levels of DNA methylation between two alleles. Both showed higher levels of DNA methylation in the maternal than paternal alleles: 37% vs 27% for ECR2 (Mann-Whitney U test, *p*-value = 0.07078) and 83% vs 64% for ECR17 (Mann-Whitney U test, *p*-value = 0.0476). These methylation level differences for ECR2 and ECR17 appeared to be significant, although the functional relevance is currently unknown. Overall, some of the ECRs maintain different levels of DNA methylation between the two alleles, supporting the possibility that this subset of ECRs may contribute to the allele-specific expression of the imprinted genes in the *Peg3* domain.

## Chromosomal conformation comparison between WT and U1/KO2

We further tested potential involvement of ECRs in the allele-specific expression of the *Peg3* domain with the Chromosomal Conformation Capture protocol (3C), in which potential interactions between promoters and enhancers can be determined for a given genomic interval [27,28]. For this series of experiments, we used the following strategy. We designed a set of oligonucleotides that are derived from the regions 100 bp-downstream of *Nco*I sites, which were then named after the adjacent ECRs (vertical blue lines in Fig 5). We also included the two primers that are derived from the promoter regions of *Peg3* and *Zim1*, which were subsequently used as anchor or base primers for PCR amplification (Zim1p and Peg3p primers). The amplification efficiency of these ECR primers with each of two base primers were tested using the template DNA that had been prepared through digestion and re-ligation of the genomic DNA isolated from the two BAC clones covering this 200-kb interval with minimal overlap (upper gel panel in Fig 5) [15]. For the actual 3C experiments, we used three sets of individual tissues—neonatal brains, adult hypothalamus and placenta—harvested from WT and U1/KO2 mice. The 3C libraries from each tissue were first analyzed with a fixed number of PCR cycles (middle and bottom gel panels in Fig 5), which were then analyzed with qPCR. To monitor the quality of each library, we also used an independent set of primers as an internal control, which is designed to detect potential interactions between the promoter and 130-kb upstream enhancer of the *Snrpn* locus (S7 and S8 Files).

The results derived from this series of 3C experiments are summarized as follows. In WT-neonatal brain samples, the overall enrichment levels of PCR products detecting potential ECR's interactions with the promoter of *Peg3* were generally greater than those with the promoter of *Zim1* (middle panels in Fig 5; blue versus red bars in Fig 6A). This may reflect the fact that the promoter activity of *Peg3* is stronger than that of *Zim1 in vivo*. The observed ECR's interaction with *Peg3* and *Zim1* are also likely allele-specific: *Peg3* on the paternal and *Zim1* on the maternal allele. This is based on the results from a set of independent 3C with the neonatal brains of KO2 pups lacking the active paternal allele of *Peg3* (Fig 1), which showed no detectable enrichment with the promoter of *Peg3*, but much greater levels of enrichment with the promoter of *Zim1* (S9 File). Also, the enrichment levels with the promoter of *Peg3* were quite variable among individual ECRs: three ECRs, ECR17, ECR18, and ECR19, showed much

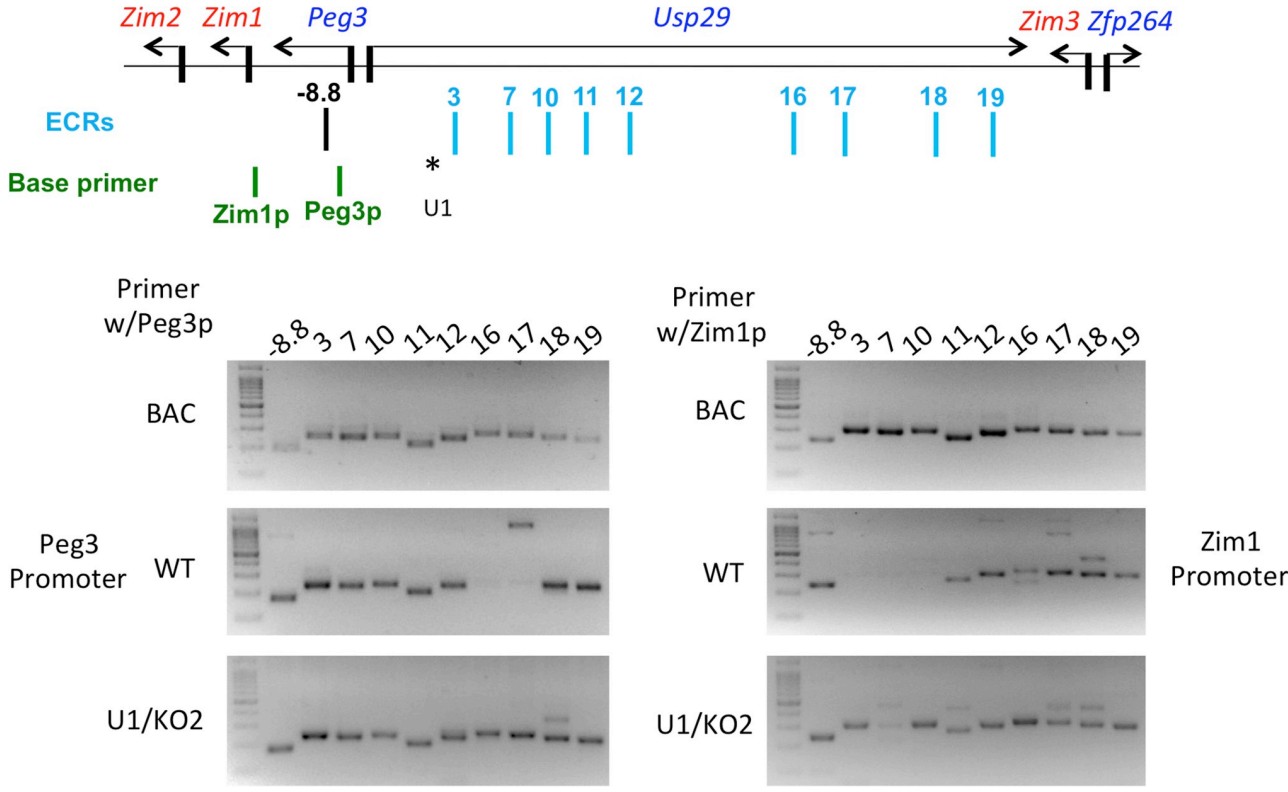

**Fig 5. 3C analyses on ECRs.** The upper panel details the genomic structure of *Peg3* domain, the relative positions of 19 ECRs and associated primers (light blue), and the two base primers (green). The position of U1 is indicated with *. The three gel images on the bottom panels represent the results from the two sets of control experiments with a BAC library, and also from the neonatal brains of wild type (WT) and the mutant with switched active alleles (U1/KO2).

greater levels of enrichment than the remaining ECRs (Fig 6A). By contrast, the enrichment levels with the promoter of *Zim1* were overall similar between individual ECRs. Finally, the overall enrichment patterns detected from the WT samples were quite different from those observed from the U1/KO2 samples (Fig 6B). It is notable that the enrichment levels at ECR17, ECR18, and ECR19 become somewhat similar between the promoters of *Peg3* and *Zim1*. On the other hand, the enrichment levels at ECR10 and ECR16 showed much greater levels of interaction with both *Peg3* and *Zim1* promoters in the U1/KO2 than in WT samples.

For each ECR, the enrichment levels of a given promoter were further compared between two alleles (Fig 7). This series of comparisons was feasible, since the two 3C libraries from the WT and U1/KO2 samples displayed a similar range of Ct values, 32.7 and 32.1, respectively, at the internal control *Snrpn* locus. According to the results, the following four types of enrichment or interaction patterns were observed among individual ECRs. The first type showed greater levels of the enrichment on the paternal allele with the promoters of both genes (**A**). This type includes ECR7, ECR11 and ECR17. The second type also displayed greater levels of enrichment on the paternal allele but only with the promoter of *Peg3* (**B**). The third type showed similar levels of the enrichment between the two alleles, thus biallelic interaction (**C**). Finally, the fourth type tended to show higher enrichment levels on the switched alleles with both promoters: the maternal allele for *Peg3* and the paternal allele for *Zim1*. This type includes ECR3, ECR10, and ECR16 (**D**). In this case, some properties associated with these ECRs might have been affected or changed by the switching of active alleles of two promoters. Thus, these

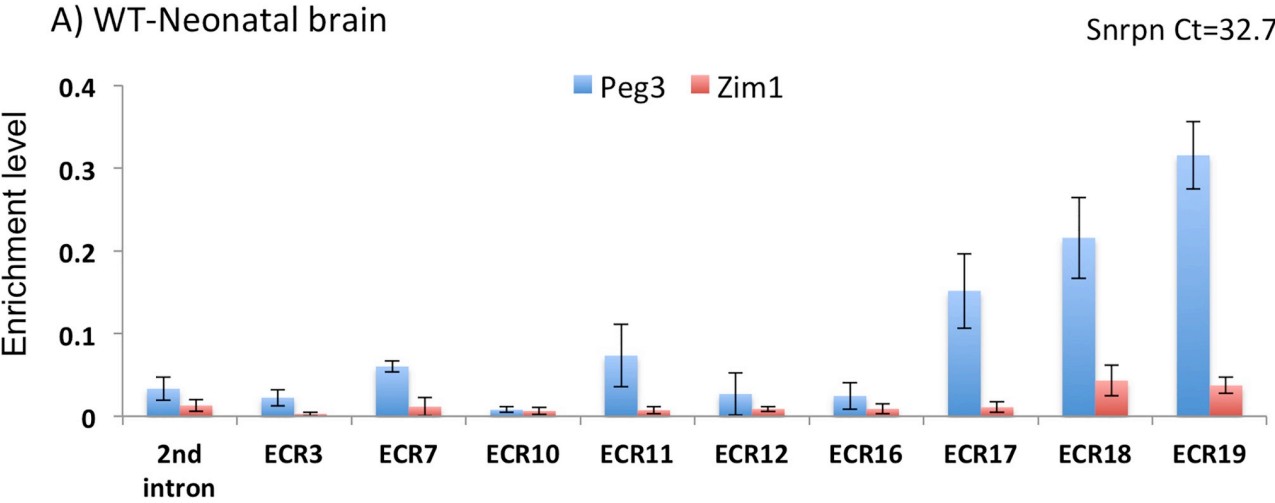

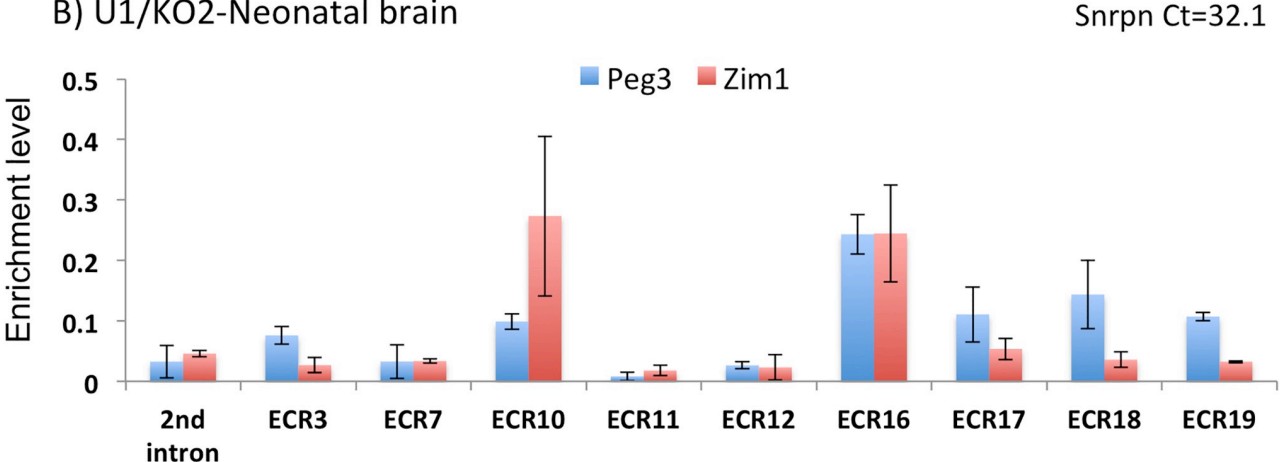

**Fig 6. qPCR analyses of 3C results.** The results from the neonatal brains of WT and U1/KO2 were further analyzed with qPCR. Relative enrichment levels of each amplicon detecting potential interaction of a given ECR with the promoters of *Peg3* (blue bars) and *Zim1* (red bars) in each library were first normalized with those derived from the BAC library. Also, successful construction of 3C libraries was monitored through measuring the enrichment level of an independent amplicon measuring the interaction between the promoter and the 130-kb upstream enhancer of *Snrpn*. The subsequent Ct values were shown upper right.

ECRs tend to show greater levels of enrichment with the newly activated promoters: *Peg3* on the maternal allele and *Zim1* on the paternal allele. On the other hand, the first two types (**A**, **B**) tend to maintain greater levels of enrichment on the paternal allele, and thus paternal-biased. These four types of the enrichment patterns were also observed from the samples prepared from the hypothalamus of 2-month-old adult mice, but not from the samples from the placentas (S5–S7 Files). In the case of 14.5-dpc (days post coitum) placenta, the interaction seemed to be mostly biallelic except the paternal-biased enrichment at three ECRs, ECR11, ECR17 and ECR18 (S6 and S7 Files). Thus, the observed four types of enrichment patterns may be tissue-specific, mostly detected in the neuronal tissues. This series of 3C analyses were repeated with two additional sets of neonatal brains, including one female set, which were shown to be reproducible. Overall, the ECRs indeed interact with the promoters of *Peg3* and

## A) Paternal-biased with *Peg3* and *Zim1*

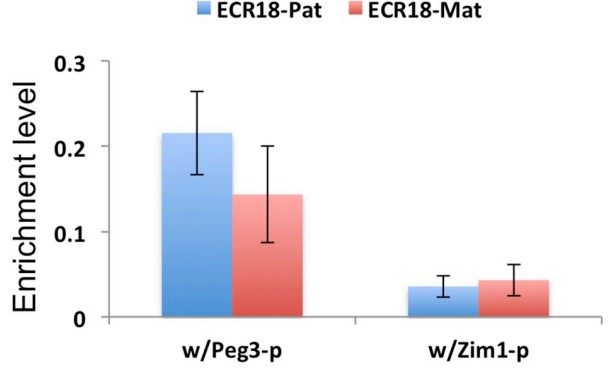

## B) Paternal-biased with *Peg3*

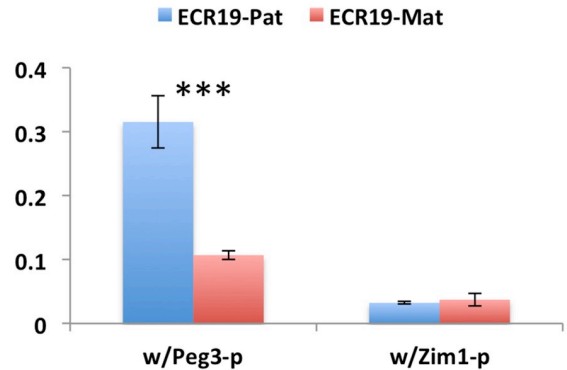

## C) Biallelic with *Peg3* and *Zim1*

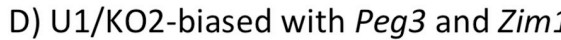

## D) U1/KO2-biased with *Peg3* and *Zim1*

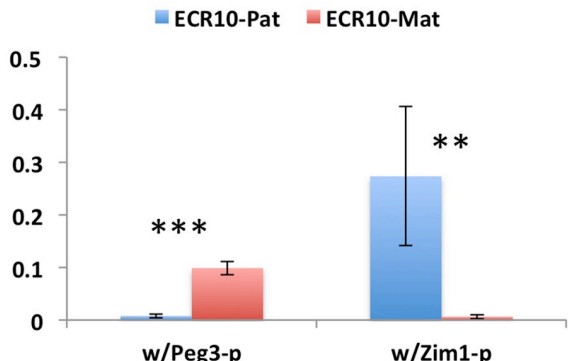

**Fig 7. Four different types of the enrichment patterns.** For a given ECR, the enrichment levels representing its potential interaction with the promoters of *Peg3* or *Zim1* were compared between two alleles, which were summarized with the average values with standard deviations. For a given promoter, the enrichment levels measured from the paternal and maternal alleles were indicated with blue and red bars. The values for the paternal and maternal alleles of *Peg3* were derived from WT and U1/KO2, respectively. On the other hand, the values for the maternal and paternal alleles of *Zim1* were derived from WT and U1/KO2, respectively. The statistical significance is indicated in the following manner: $^{*}$, $p < 0.05$; $^{**}$, $p < 0.01$; and $^{***}$, $p < 0.001$.

*Zim1*, and also a subset of ECRs, including ECR7, ECR11, ECR17 and ECR19, tend to interact better or more frequent on the paternal than maternal alleles.

## Discussion

In the current study, we compared the expression levels and patterns of *Peg3* and *Zim1* in WT mice and U1/KO2 animals with switched active alleles. We found that the expression levels of *Peg3* and *Zim1* were altered in the U1/KO2 mice, suggesting functional non-equivalence between the two alleles. Follow-up studies further identified ECRs localized upstream of *Peg3* that are allele-biased, in terms of both DNA methylation levels and interactions with the promoters of *Peg3* and *Zim1* (Fig 8). Thus, this suggests that some long-distance enhancers also contribute to the allelic expression of genes within the *Peg3* domain, in addition to known DMRs and the promoters.

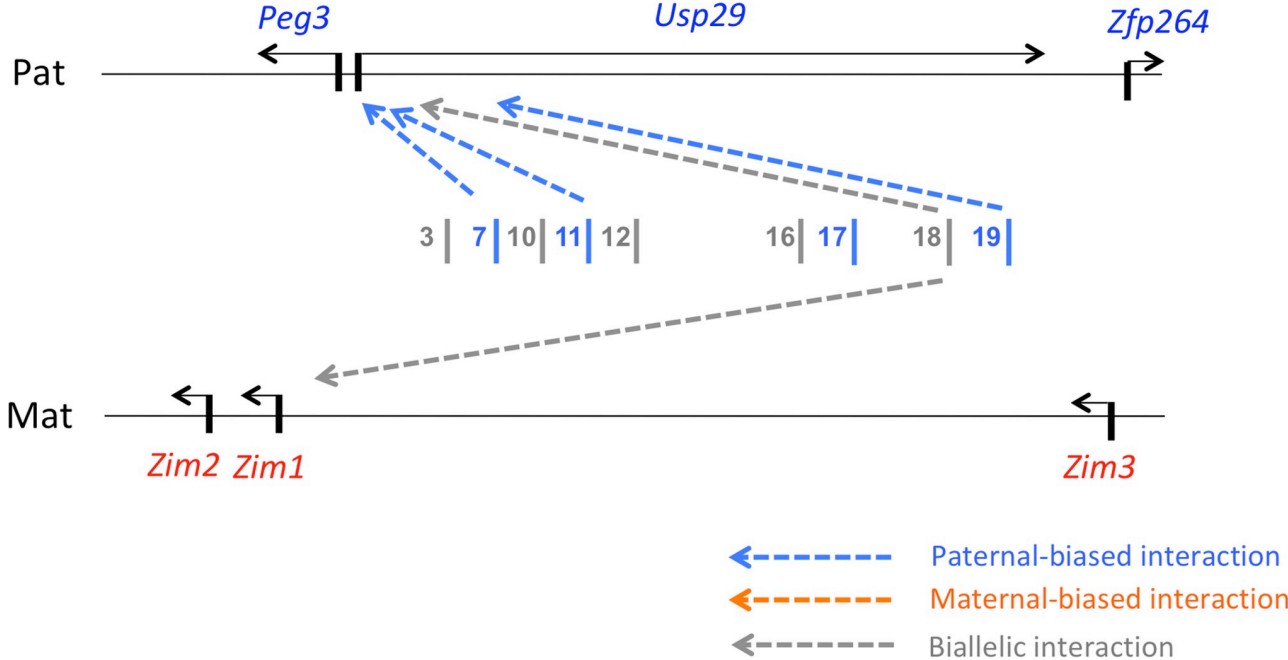

**Fig 8. Schematic representation of allele-biased features of ECRs.** The 500-kb genomic interval of the *Peg3* domain harbors paternally expressed *Peg3*, *Usp29*, *Zfp264* (blue) and maternally expressed *Zim1*, *Zim2*, *Zim3* (red). The middle 200-kb region is filled with 19 ECRs that are putative enhancers for the transcription of the *Peg3* domain. This schematic diagram summarizes the results from 3C results, revealing the presence of several allele-biased ECRs, including ECR7, ECR11, ECR17, and ECR19: the interaction of these ECRs with the promoters of *Peg3* and *Zim1* is stronger or more frequent on the paternal than maternal allele, thus paternal-biased. This may provide an explanation for the differences observed between WT and U1/KO2 with switched active alleles. Switching the active alleles of *Peg3* and *Zim1* may disrupt the allele-specific interactions that are already programmed between the promoters and these ECRs, thus causing some effects on the expression levels of the *Peg3* domain.

The results from expression profiling demonstrated that two imprinted genes, *Peg3* and *Zim1*, showed clear differences between two alleles (Fig 2). The differences were more obvious within the three regions of adult brain, including the hypothalamus, olfactory bulbs and cerebellum. Interestingly, the expression level changes were not unidirectional among these brain regions: the expression levels of *Peg3* and *Zim1* were slightly greater in the hypothalamus, but lower in the olfactory bulbs and cerebellum of the U1/KO2 animals (Fig 2 **and** S1 File). On the other hand, two independent series of immunostaining experiments demonstrated that the spatial expression patterns of *Peg3* and *Zim1* within the hypothalamus were overall similar between WT and U1/KO2 (Fig 3), suggesting that the switching of active alleles does not interfere with transcriptional programs directing the spatial expression patterns of *Peg3* and *Zim1*. This finding further suggests that the expression level differences occur within the same set of cells, but not from ectopic expression of *Peg3* and *Zim1* in U1/KO2 mice. Nonetheless, we cannot rule out the possibility that there may be some subtle differences in the spatial expression patterns of *Peg3* and *Zim1* in U1/KO2 mice, but in a very cell-type-specific manner within the hypothalamus. This is further supported by the fact that the expression levels of the hormonal genes, *Oxt*, *Avp*, and *Ghrh*, of the hypothalamus exhibited differences between WT and U1/KO2 (S3 File). The latter gene in particular is consistent with the previous observation that the switching resulted in the increased and decreased body weights among the females and males, respectively, at weaning age [18], indicating effects of the switching on growth hormone pathways. Although unlikely, however, we cannot rule out another possibility that some of the observed differences might be caused by some unknown effects of the two mutant alleles, U1 and KO2, that have been used for the switching. Overall, the switching of active alleles

appeared to cause clear changes in the expression levels of *Peg3* and *Zim1*, and suggested that some regions other than the promoters of *Peg3* and *Zim1* might differ in activity and function between two alleles.

In particular, our results show that several ECRs localized upstream of *Peg3* do indeed differ in terms of DNA methylation and promoter interaction between the two alleles, and that their altered activity may be responsible for the observed differences between WT and U1/KO2 (Figs 4–7). DNA methylation analyses indicated that two ECRs, ECR2 and ECR17, exhibited slightly lower levels of DNA methylation on the paternal than maternal allele (Fig 4). Furthermore, 3C experiments demonstrated that the interaction of four ECRs—ECR7, ECR11, ECR17 and ECR19—with the promoters of *Peg3* and *Zim1* tend to be stronger or more frequent on the paternal than maternal allele (Fig 7). This suggests that some unknown properties associated with these ECRs may be biased toward the paternal allele (Fig 8). The molecular nature of these paternal-biased properties is currently unknown; however, the bias is likely to be due to epigenetic modifications, since the two alleles of these ECRs are identical in terms of DNA sequence. Consistent with this prediction, these ECRs are indeed associated with two histone marks, H3K4me1 and H3K27ac [15,16]. According to recent studies involving ATAC-seq, the open chromatin structure of some of these ECRs turns out to be also allele-biased in the neural precursor cells: maternal-biased open structure for ECR2 and ECR17 and paternal-biased open structure for ECR19 [29]. Interestingly, the pioneer factor MyoD, a well-known E-box-binding protein, has been shown to bind to a subset of ECRs, yet the list of the target sites identified within the *Peg3* domain overlaps quite well with the three ECRs, including ECR7, ECR11, ECR17 [16]. This overlap may not be a simple coincidence, but rather a strong hint suggesting some shared features among these ECRs. In this case, the paternal biased properties described in this study may be one of these shared features. Overall, the identification of the paternal-biased ECRs from the *Peg3* domain is a novel and significant finding, opening door to new insights regarding potential mechanisms for the monoallelic or allele-biased expression patterns that are frequently associated with mammalian genomes [30,31].

## Materials and methods

### Ethics statement

All the experiments related to mice were performed in accordance with National Institutes of Health guidelines for care and use of animals, and also approved by the Louisiana State University Institutional Animal Care and Use Committee (IACUC), protocol #16–060.

### Mouse breeding

In the current study, we used two mutant strains that have been previously characterized, $Peg3^{KO2/+}$ and $Peg3^{U1/+}$ strains [12,14]. Female heterozygotes for $Peg3^{U1/+}$ were crossed with male heterozygotes for $Peg3^{KO2/+}$. The subsequent pups were analyzed in terms of sex and genotype. For genotyping, genomic DNA was isolated from either clipped ears or tail snips by incubating the tissues overnight at 55°C in the lysis buffer (0.1 M Tris-Cl, pH 8.8, 5 mM EDTA, pH 8.0, 0.2% SDS, 0.2 M NaCl, 20 μg/ml Proteinase K). The isolated DNA was subsequently genotyped using the following set of primers: for the KO2 allele, Primer A (5′-TGA CAAGTGGGCTTGCTGCAG-3′), B (5′-GGATGTAAGATGGAGGCACTGT-3′), and D (5′-AGGGGAGAACAGACTACAGA-3′); for the U1 allele, P1 (5′-TAGCAAGGGAGAGGGCCTAG-3′), P2 (5′-GGAAGCCTCCATCCGTTTGT-3′), and P3 (5′-AGCACAGCTAGAAATACAC AGA-3′). The sex of the pups was determined through PCR using the following primer set: mSry-F (5′-GTCCCGTGGTGAGAGGCACAAG-3′) and mSry-R (5′-GCAGCTCTACTCCAG TCTTGCC-3′).

## RNA isolation, cDNA synthesis, and qRT-PCR analyses

Total RNA was isolated from the various tissues of adult mice using a commercial kit (Trizol, Invitrogen) according to manufacturer's instructions. The total RNA was reverse-transcribed using the M-MuLV kit (Invitrogen), and the subsequent cDNA was used as a template for quantitative real-time PCR. This analysis was performed with the iQ SYBR green supermix (Bio-Rad) using the ViiA™ 7 Real-Time PCR System (Life Technologies). All qRT-PCR reactions were carried out for 40 cycles under standard PCR conditions. The analyses of the results from qRT-PCR were described previously [32]. Statistical significance of potential difference in expression levels of a given gene between two samples was tested with Mann-Whitney U test (https://www.socscistatistics.com/tests/mannwhitney/default2.aspx). The information regarding individual primer sequences is available as S7 File.

## Immunohistochemistry

Each mouse was anesthetized with an intraperitoneal injection of a ketamine/xylazine cocktail (87.5 mg/kg ketamine; 12.5 mg/kg xylazine) at a dosage of 0.1 ml per 20-gram body weight. The animals were then transcardially perfused with 0.1 M sodium phosphate-buffered saline (PBS: pH 7.2–7.4) and fixed with 4% paraformaldehyde in 0.1 M phosphate buffer (PB: pH 7.2–7.4). Mice were decapitated, and heads were post-fixed overnight in the same fixative. Coronal sections were transected from the hypothalamus at 40 μm thickness using a vibratome (Leica VT1200 S, Leica, Mannheim, Germany), and placed in PBS containing 0.5% Triton X-100 (PBST). The free-floating brain sections were incubated with an in-house primary antibody against ZIM1 and the primary PS38 antibody against oxytocin-neurophysin (provided by H. Gainer, NIH) at dilutions of 1:1000 and 1:500, respectively. This incubation was carried out in PBST for 48–72 hours at 4˚C with continuous gentle agitation. Sections were washed three times with fresh PBST, followed by incubation with goat anti-rabbit antibody conjugated with DyLight 488 (Jackson ImmunoResearch, West Grove, PA) and goat antibody conjugated with DyLight 649 (Jackson ImmunoResearch, West Grove, PA). Both incubations were performed at 1:400 dilution in PBST overnight. The sections were washed three times with PBST, mounted on slides, and cover-slipped with an anti-fading agent that consists of 4.8g PVA, 12g glycerol, 12 mL dH$_2$O, 24 mL 0.2M Tris-HCl, and 1.25g DABCO (1,4-diazabicyclo[2.2.2] octane). Fluorescence images were acquired digitally (Eclipse 80i equipped with a digital camera, DS-QiMc, Nikon, Tokyo, Japan). ImageJ software was used to process the images in dynamic range with minimal alterations.

## DNA methylation analysis

DNA was first isolated from the brains of one-day-old neonates of the F1 hybrid that had been derived from the crossing of female C57BL/6J and male PWD/PhJ. The isolated DNA was treated with the bisulfite conversion protocol [25,26]. The converted DNA was subsequently used as a template for PCR reactions targeting each ECR. The amplified products were individually cloned into a commercial vector, and on average 16 clones for each PCR product were sequenced for its final DNA methylation level. The information regarding the sequences of oligonucleotides for each ECR is available as S7 File.

## 3C (Chromatin Conformation Capture)

The 3C method was performed as detailed in the Current Protocols in Molecular Biology Handbook unit 21.11 in Supplement 74 [27,28]. In short, the two mouse BAC (bacterial artificial chromosome) clones, RP23-178C5 (Invitrogen) and RP23-117K9 (CHORI), were used for

generating the control template libraries. These BACs cover the majority of the *Peg3* domain (nucleotide positions 6,610,343–6,929,458 in mouse chromosome 7) with an approximately 9,000 bp overlap (6,793,948–6,802,969). The purified DNA from these two BACs (total 10 μg with an equal ratio) was digested with *Nco*I, religated, and finally prepared for the control template libraries. For the actual 3C experiments, we harvested the following three tissues of WT and U1/KO2 mice: the one-day-old neonatal brains, the hypothalamus of 2-month-old adult mice, and the 14.5-d.p.c. placentas. Each tissue was homogenized in PBS, crosslinked with 1% formaldehyde for 10 minutes, and finally divided into several fractions at the concentration of 10 mg per aliquot. Each aliquot was used for *Nco*I digestion, religation, and DNA purification.

For PCR analysis, each primer was designed from a region 100-bp downstream of a given *Nco*I site. The efficiency and compatibility of a given primer set were tested using serial dilutions of the control template libraries that had been prepared from the two BAC clones. For an initial survey, a fixed number of PCR cycles was performed using each base primer along with a panel of ECR primers. For qPCR analysis, 1 μl of either the control or the 3C libraries from mouse tissues was used as a template with SYBR Green Premix reagents (BioRad). The parameters for PCR are as follows: 95°C for 4 mins, 40 repetitions of the following cycle of 95°C for 15 sec, 65°C for 30 sec, 72°C for 30 sec. A camera capture setting was included after the 65°C step to monitor the formation of PCR product. A melt curve step ranging from 55–95°C with a hold of 10 sec and a temperature increment of 0.5°C was included at the end of the PCR to monitor the quality of PCR products.

## Supporting information

**S1 File. This file contains the summary of the expression level comparison of *Peg3* and *Zim1* in the cortex (A) and cerebellum (B) between WT and U1/KO2.**
(TIF)

**S2 File. This file contains the summary of the expression level comparison of *Peg3* and *Zim1* in the kidney (A) and gonadal fat (B) between WT and U1/KO2.**
(TIF)

**S3 File. This file contains the summary of the expression level comparison of *Avp* (A), *Oxt* (B), *Oxtr* (C), and *Ghrh* (D) in the hypothalamus between WT and U1/KO2.**
(TIF)

**S4 File. This file contains the magnified view of the PVN areas of the hypothalamus that have been immunostained with anti-ZIM1 antibody.**
(TIF)

**S5 File. This file contains the summary of the 3C results that have been derived from the hypothalamus of 2-month-old adult mice of WT (A) and U1/KO2 (B).**
(TIF)

**S6 File. This file contains the summary of the 3C results that have been derived from the 14.5-dpc placentas of WT (A) and U1/KO2 (B).**
(TIF)

**S7 File. This file contains the compiled raw data sets that have been derived from qRT-PCR-based expression surveys and also from qPCR-based 3C analyses.** This file also contains the information regarding all the oligonucleotides used for expression, DNA methylation and 3C analyses.
(XLSX)

**S8 File. This file contains the 3C results demonstrating the interaction of the promoters of** *Snrpn* **and** *Ube3a* **with several ECRs that are localized upstream of** *Snrpn***.**
(TIF)

**S9 File. This file contains the summary of the 3C results that have been derived from the mutant animals with the paternal deletion of KO2, demonstrating the allele-specific interaction of the promoter of** *Peg3* **with ECRs.**
(TIF)

## Acknowledgments

We would like to thank Drs. Jin Xu and Howard Chang at Stanford University for sharing ATAC-seq data.

## Author Contributions

**Conceptualization:** Joomyeong Kim.

**Data curation:** Joomyeong Kim, Wesley D. Frey, Kaustubh Sharma, Subash Ghimire.

**Formal analysis:** Joomyeong Kim, Wesley D. Frey, Kaustubh Sharma, Subash Ghimire, Ryoichi Teruyama.

**Funding acquisition:** Joomyeong Kim.

**Investigation:** Joomyeong Kim.

**Methodology:** Joomyeong Kim, Wesley D. Frey, Kaustubh Sharma, Subash Ghimire, Ryoichi Teruyama.

**Project administration:** Joomyeong Kim.

**Resources:** Joomyeong Kim, Lisa Stubbs.

**Supervision:** Joomyeong Kim.

**Validation:** Joomyeong Kim.

**Visualization:** Joomyeong Kim.

**Writing – original draft:** Joomyeong Kim, Lisa Stubbs.

**Writing – review & editing:** Joomyeong Kim.

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
