## [Decision Letter · Decision Letter 0]

6 Aug 2019

PONE-D-19-17945

Allele-specific enhancer interaction at the Peg3 imprinted domain

PLOS ONE

Dear Dr. Kim,

Thank you for submitting your manuscript to PLOS ONE. After careful consideration, we feel that it has merit but does not fully meet PLOS ONE’s publication criteria as it currently stands. Therefore, we invite you to submit a revised version of the manuscript that addresses the points raised during the review process.

Although the referees all thought this work was interesting, they each had criticisms of the manuscript that you need to address (please address all the comments below). 

In particular:

1, Please take care in your conclusions, avoiding over-interpreting subtle differences.

2. Clarify the statistics to make sure that you are considering differences between biological replicates, not between technical replicates.

We would appreciate receiving your revised manuscript by Sep 20 2019 11:59PM. To enhance the reproducibility of your results, we recommend that if applicable you deposit your laboratory protocols in protocols.io, where a protocol can be assigned its own identifier (DOI) such that it can be cited independently in the future. For instructions see: http://journals.plos.org/plosone/s/submission-guidelines#loc-laboratory-protocols

We look forward to receiving your revised manuscript.

Kind regards,

Keith William Brown, Ph.D.

Academic Editor

PLOS ONE

Journal Requirements:

1. Thank you for including your funding statement; "The funders had no role in study design, data collection and analysis, decision to publish, or preparation of the manuscript."

Please provide an amended Funding Statement that declares *all* the funding or sources of support received during this specific study (whether external or internal to your organization) as detailed online in our guide for authors at http://journals.plos.org/plosone/s/submit-now.  

Please state what role the funders took in the study.  If any authors received a salary from any of your funders, please state which authors and which funder. If the funders had no role, please state: "The funders had no role in study design, data collection and analysis, decision to publish, or preparation of the manuscript."

Reviewers' comments:

Reviewer's Responses to Questions

**Comments to the Author**

1. Is the manuscript technically sound, and do the data support the conclusions?

Reviewer #1: Partly

Reviewer #2: No

Reviewer #3: Partly

2. Has the statistical analysis been performed appropriately and rigorously? 

Reviewer #1: Yes

Reviewer #2: No

Reviewer #3: No

3. Have the authors made all data underlying the findings in their manuscript fully available?

Reviewer #1: Yes

Reviewer #2: Yes

Reviewer #3: No

4. Is the manuscript presented in an intelligible fashion and written in standard English?

Reviewer #1: Yes

Reviewer #2: No

Reviewer #3: No

5. Review Comments to the Author

Reviewer #1: This is a novel study of gene regulation at the imprinted Peg3 locus using mouse KO lines that have been described in several previous papers from the same group. The mice harbour two different mutations that together switch allelic expression of imprinted genes on the maternal and paternal alleles. Here, by comparing expression, methylation and enhancer/promoter interactions they infer quantitative differences may exist in the interactions between the promoters and distal enhancer regions that are allele-specific.

The data generally look sound and my main advice is that the authors exercise greater caution in the way they choose to interpret their findings. Although they show some quantitative differences to be statistically significant they are relatively subtle. While these differences might be due to allele-specific in promoter/enhancer interactions involving epigenetic changes this remains to be proven. The observed differences could also be a direct consequence of the two different gross changes that have been introduced to the mutant loci. Thus, I would like to see a clear caveat to this effect included in the discussion

Some specific points:

Introduction (p3, para 1). Is it true to say “The majority of imprinted genes are found only in placental mammals”?

Introduction (p3, para 1). The sentence beginning “The mechanisms through imprinted…” does not make grammatical sense.

Introduction (p3, para 2). In what context is the Peg3 domain “well conserved”, do you mean specifically between mouse and human?

Introduction (p3, para 2). Reference to Figure 1 would be helpful.

Results (p5, para 1). The sentence beginning “The schematic representations…” belongs in the appropriate figure legend.

Results (p6, para 2). The last line, beginning “on the other hand..” should perhaps refer to expression ‘differences’ being ‘indistiguishable’ rather than ‘levels’ being ‘inconclusive’.

Results (p6, para 3). Make clear you are talking specifically about male kidney and male gonadal fat.

Results (p6, para 3). In the sentence beginning “Although the observed differences..” it is not clear whether the comment “at least ranging from 10 to 100 fold…” refers to brain or non-neural tissues. The whole sentence needs a rethink.

Results (p7, para 1). “Fourth we analysed the exp[ression levels…” in which tissues etc?

Results (p7, para 2). “The results from this series…” Remove this unnecessary statement and others that are similar elsewhere.

Results (p7, para 2). You state that Peg3 expression is ubiquitous in adult brain then later that Zim1 is uniquely expressed in ependymal cells; one of these statements must be wrong.

Results (p8, para 1). You note that Zim1 expression is “very similar” to Peg3 in PVN and SON. If Peg3 expression is ubiquitous and Zim1 expression more restricted, ZIm1 expression must form a sub-set of Peg3 expression and I don’t see how this forms good evidence they “share long-distance enhancers”. In addition, you specifically indicate enhancers “localized in the middle 200-kb region” of the domain but this cannot be inferred from expression patterns alone.

Results (p8, para 2). Why give a DNA methylation range (“8-37%” for enhancers with low levels but not for those with high methylation levels? You also state that maternal versus paternal differences “appeared to be significant”, but on what basis (statistical test)?

Results (p9, para 2). Not clear that BAC DNA provides a good control for amplification efficiency for 3C experiments involving native cellular DNA.

Results (p9, para 1). I can’t find any primer details for the 3C experiments. Also, why was the Snrpn control chosen, has it been validated previously, perhaps in a prior publication?

Results (p10, para 1 and 2). Some potentially important data on Zim1 is ‘not shown’ and there is a lack of quantification until Figure 7. It is not clear to me how this comparison has been done, including the stats, and I am struggling to evaluate its validity.

Discussion (p12, par 2). Four ECRs are indicated (ECR7, 11, 17 and 19), which is different to the 3 highlighted on page 10 of the Results (ECR17, 18 and 19).

Discussion (p13, par 1). The section beginning “Interestingly, the pioneer factor MyoD…” and ending with the sentence “Further, this suggests…” comes across as a rather clumsy piece of speculation. I would consider a careful rewrite or omitting this entirely.

Reviewer #2: In the manuscript entitled “Allele-specific enhancer interaction at the Peg3 imprinted domain” by Kim et al. identified putative enhancers localized upstream of Peg3 that displayed allele-biased DNA methylation, and they suggested that some of them participate in allele-biased chromosomal conformations with regional Peg3 and Zim1 promoters and that such long-distance enhancers may contribute to allelic expression of Peg3 and Zim1. In the previous report, the authors reported their interesting double KO mice with switched active alleles of Peg3 and Zim1, that is maternally expressed Peg3 and paternally expressed Zim1 by deletion of both Peg3-DMR (ICR) and oocyte-specific alternative promoter (U1) of Peg3. I agree that their switched mouse is a very good tool for elucidating imprinting mechanism and identification of the paternal-biased ECRs from the Peg3 domain is also a novel and very important finding, however, I don’t think that their results actually support their main conclusion (suggestion) that the long-distance enhancers may contribute to allelic expression of Peg3 and Zim1.

Major comments

1. Imprinted domains are very unique because expression profiles of both of the paternal and maternal alleles are different. Therefore, the authors need to carry out 3C experiment on paternal and maternal alleles separately. The authors insisted that the observed ECR’s interaction with Peg3 and Zim1 are likely allele-specific: Peg3 on the paternal and Zim1 on the maternal allele. However, it is not guaranteed.

In the U1/KO2 mice with maternally expressed Peg3 and paternally expressed Zim1, the interaction of ECR’s and Peg3/Zim1 promoters (Fig. 6B) is far from the reversed version of wild type (Fig. 6A), suggesting that the pattern of interaction between the ECR’s and Peg3/Zim1 promoters is more complex and represents mixed pattern of both of the paternal and maternal interaction. Therefore, they should present the interaction profiles of paternal and maternal alleles in the U1/KO2 case, separately.

2. The authors suggested that the long-distance enhancers may contribute to allelic expression of Peg3 and Zim1, however, which ones contribute to the change of Peg3/Zim1 allelic expression although I agree that the expression levels of Peg3 and Zim1 are affected by some of the ECR’s by modifying chromosome conformation around Peg3 and Zim1 promoters?

3. DNA methylation analysis was carried out only using wild type mice but not using U1/KO2 mice. However, it is very important to know the DNA methylation status of the ECR2 and ECR17 in U1/KO2 mice because their methylation difference may cause change of interaction between these ECRs and Peg3/Zim1 promoters.

4. I don’t think that interaction between ECRs and Peg3/Zim1 promoter in Fig. 8 reflects the authors experimental data and their explanation in the text. For one example, why do the fourth type (ECR3, ECR10 and ECR16) tended to show higher enrichment levels on the switched alleles with both promoter (the maternal allele for Peg3 and the paternal allele for Zim1) is classified by biallelic interaction because no enrichment was observed in the wild type (Fig. 6).

Minor comments

1. Although it is necessary to explain the generation of a mouse mutant (U1/KO2), Figure 1 may be redundant because it’s the same as the previous report (ref. 18). The authors should include the information of the position of U1 promoter in the ECR region.

2. In Fig. 4 (on DNA methylation levels of ECRs), authors described that ECR2 and ECR17 exhibited different levels of DNA methylation between paternal and maternal alleles. However, it is difficult to say that the difference of methylation levels are significant between two alleles because it was provided the data of one sample.

3. In Figure 3, images of immunostaining are low resolution.

Reviewer #3: Allele-specific enhancer interaction at the Peg3 imprinted domain

Kim et al.

The authors look at the role of ECRs in the control of imprinted gene expression in the Peg3 domain.

They also investigate the consequences of a double switch so the paternal chromosome behaves maternally

and the maternal chromosome behaves paternally.

Major comments.

For the gene expression data shown in Figure 1 only 8 animals appear to have been studied, this is far too few to draw

conclusions from and perform meaningful statistical analyses. The error bars shown are for technical replicates - I expect on biological replicates they would be much greater.

Are the wildtype samples littermates of the mutants? The expression levels of mutants should be normalised to wildtype litter mates to ensure they had the same in utero environment and they are exactly the same age.

Also, it is not clear if the genetic background of the chromosome that each deletion is on is the same. Many knockouts are derived on non-C57B/6 strains and the results seen could purely be down to genetic background differences.

For the methylation analyses the authors used C57BL/6J females crossed with PWD/PhJ males. However,

for robust methylation analysis reciprocal crosses should be used to eliminate any genetic background effects.

In the 3C analysis the authors do not explain how they are able to distinguish the maternal and paternal alleles. Is this through SNPs or are the base primers located within the deletion so they are only interrogating interactions

on one chromosome? As this provides a major part of their discussion their methods should be described more

fully.

In Figure 5, the middle panel for the Peg3 promoter wild-type - the band for ECR17 is much larger than expected - why is this?

They mention a further set of 3Cs in KO2 pups lacking the paternal Peg3 allele - these data should be in the supplementary material.

The authors mention a that the 3C analyses were repeated twice. The data from all three biological replicates should be combined for the main figures rather that showing them separately. This would allow the statistical analyses to be performed on biological replicates rather than technical replicates (as I assume are being shown in Figure 6).

Minor comments

What do the authors mean by "placental mammals". To make it clearer authors should use the accepted

terminology:

Eutherians - mammals that are not marsupials or monotremes

Therians - eutherian and marsupials

Line 5 should read eutherian evolution - as only 5 genes are known to be imprintied in marsupials

Line 9 should read the majority of imprinted genes are found only in eutherian mammals.

Line 10 should read imprinting may have evolved (or emerged) in the therian lineage. - A biological process can not be invented>

Line 11 the sentence beginning "The mechanisms through imprinting " needs re-wording as it does not make sense.

In paragraph 3 of the introduction the authors should say that the ECRs are located within the Usp29 transcript.

The orientation of the region in figures switches from figure to figure which make it difficult for a non-expert in this region

to follow what is going on.

6. PLOS authors have the option to publish the peer review history of their article (what does this mean?). If published, this will include your full peer review and any attached files.

Reviewer #1: No

Reviewer #2: No

Reviewer #3: No

---

## [Author Response · Author response to Decision Letter 0]

17 Sep 2019

Response to Review Comments

Reviewer #1: This is a novel study of gene regulation at the imprinted Peg3 locus using mouse KO lines that have been described in several previous papers from the same group. The mice harbour two different mutations that together switch allelic expression of imprinted genes on the maternal and paternal alleles. Here, by comparing expression, methylation and enhancer/promoter interactions they infer quantitative differences may exist in the interactions between the promoters and distal enhancer regions that are allele-specific.

The data generally look sound and my main advice is that the authors exercise greater caution in the way they choose to interpret their findings. Although they show some quantitative differences to be statistically significant they are relatively subtle. While these differences might be due to allele-specific in promoter/enhancer interactions involving epigenetic changes this remains to be proven. The observed differences could also be a direct consequence of the two different gross changes that have been introduced to the mutant loci. Thus, I would like to see a clear caveat to this effect included in the discussion

------- Response: We have included a sentence describing this potential caveat of the current study in the second paragraph of the Discussion section.

Some specific points:

Introduction (p3, para 1). Is it true to say “The majority of imprinted genes are found only in placental mammals”?

-----Response: Yes, that is correct.

Introduction (p3, para 1). The sentence beginning “The mechanisms through imprinted…” does not make grammatical sense.

------- Response: We have removed that sentence.

Introduction (p3, para 2). In what context is the Peg3 domain “well conserved”, do you mean specifically between mouse and human?

------- Response: The Peg3 domain is well conserved among several mammals, including human, mouse, sheep, dog, and cow.

Introduction (p3, para 2). Reference to Figure 1 would be helpful.

------- Response: We have added a reference pointer to Fig 1 in that paragraph.

Results (p5, para 1). The sentence beginning “The schematic representations…” belongs in the appropriate figure legend.

------- Response: We have moved this sentence to the legend of Fig 1.

Results (p6, para 2). The last line, beginning “on the other hand..” should perhaps refer to expression ‘differences’ being ‘indistinguishable’ rather than ‘levels’ being ‘inconclusive’.

------- Response: We have corrected this as suggested.

Results (p6, para 3). Make clear you are talking specifically about male kidney and male gonadal fat.

------- Response: We have added the term 'male' in the two sentences.

Results (p6, para 3). In the sentence beginning “Although the observed differences..” it is not clear whether the comment “at least ranging from 10 to 100 fold…” refers to brain or non-neural tissues. The whole sentence needs a rethink.

------- Response: We have further clarified this by modifying the sentence.

Results (p7, para 1). “Fourth we analysed the expression levels…” in which tissues etc?

------- Response: We have added the phrase 'in the hypothalamus set.'

Results (p7, para 2). “The results from this series…” Remove this unnecessary statement and others that are similar elsewhere.

------- Response: We put a summary statement at the end of the corresponding paragraph to emphasize the main point of each experiments. We have modified several summary sentences not to be redundant.

Results (p7, para 2). You state that Peg3 expression is ubiquitous in adult brain then later that Zim1 is uniquely expressed in ependymal cells; one of these statements must be wrong.

------- Response: Peg3 is expressed in the majority of neuron cells in the hypothalamus, whereas the expression of Zim1 was more limited to the small areas, such as PVN and SON. Also, we detected high expression levels of Zim1 in the ependymal cells from the current study. Overall, we do not feel these statements are contracting to each other.

Results (p8, para 1). You note that Zim1 expression is “very similar” to Peg3 in PVN and SON. If Peg3 expression is ubiquitous and Zim1 expression more restricted, ZIm1 expression must form a sub-set of Peg3 expression and I don’t see how this forms good evidence they “share long-distance enhancers”. In addition, you specifically indicate enhancers “localized in the middle 200-kb region” of the domain but this cannot be inferred from expression patterns alone.

------- Response: We agree with the reviewer on this point that we need to have additional evidence to support the statements. However, we also strongly believe that the following scenario is most likely. Although Peg3 expression is more ubiquitous than Zim1 expression in the hypothalamus, both genes share a similar spatial expression pattern, the expression within the PVN and SON areas. This unique spatial expression pattern between these two genes is a strong indication that they share a set of cis-regulatory elements, or enhancers. Further, according to the surveys of the ENCODE dataset and also our own results (Kim and Ye 2016), the middle 200-kb region has the majority of potential enhancers within this imprinted domain. Thus, we believe that the shared enhancers are likely localized within the middle 200-kb region.

Results (p8, para 2). Why give a DNA methylation range (“8-37%” for enhancers with low levels but not for those with high methylation levels? You also state that maternal versus paternal differences “appeared to be significant”, but on what basis (statistical test)?

------- Response: We have included the methylation level range of another set of ECRs with high methylation levels. We have also included the p values derived from statistical analyses in the text.

Results (p9, para 2). Not clear that BAC DNA provides a good control for amplification efficiency for 3C experiments involving native cellular DNA.

------- Response: Purified BAC DNA goes through a series of similar digestion and ligation, which is subsequently used for PCR amplification to test the feasibility of a given 3C scheme as well as the amplification efficiency of each amplicon. We believe that this is a standard practice for 3C experiments. However, we also agree with the reviewer that this may not be an ideal control for native cellular DNA. 

Results (p9, para 1). I can’t find any primer details for the 3C experiments. Also, why was the Snrpn control chosen, has it been validated previously, perhaps in a prior publication?

------- Response: We are the first group reporting this particular 3C on the Snrpn locus. The information regarding the primers has been included in Supporting information 7. We are also providing an additional set of the detailed information regarding this series of 3C as Supporting information 8.

Results (p10, para 1 and 2). Some potentially important data on Zim1 is ‘not shown’ and there is a lack of quantification until Figure 7. It is not clear to me how this comparison has been done, including the stats, and I am struggling to evaluate its validity.

------- Response: This set of results has been included as Supporting information 9. The results in Fig 7 have been derived by comparing the relative enrichment values of each amplicon or interaction between WT and U1/KO2 animals. The detailed information regarding the raw vales and stats have been included as Supporting information 7. 

Discussion (p12, par 2). Four ECRs are indicated (ECR7, 11, 17 and 19), which is different to the 3 highlighted on page 10 of the Results (ECR17, 18 and 19).

------- Response: The ECR18 was not allele-biased but is known to be a target site for MyoD, thus has been included in this statement. We have corrected this mistake.

Discussion (p13, par 1). The section beginning “Interestingly, the pioneer factor MyoD…” and ending with the sentence “Further, this suggests…” comes across as a rather clumsy piece of speculation. I would consider a careful rewrite or omitting this entirely.

------- Response: We have removed this speculative sentence along with two references.

Reviewer #2: In the manuscript entitled “Allele-specific enhancer interaction at the Peg3 imprinted domain” by Kim et al. identified putative enhancers localized upstream of Peg3 that displayed allele-biased DNA methylation, and they suggested that some of them participate in allele-biased chromosomal conformations with regional Peg3 and Zim1 promoters and that such long-distance enhancers may contribute to allelic expression of Peg3 and Zim1. In the previous report, the authors reported their interesting double KO mice with switched active alleles of Peg3 and Zim1, that is maternally expressed Peg3 and paternally expressed Zim1 by deletion of both Peg3-DMR (ICR) and oocyte-specific alternative promoter (U1) of Peg3. I agree that their switched mouse is a very good tool for elucidating imprinting mechanism and identification of the paternal-biased ECRs from the Peg3 domain is also a novel and very important finding, however, I don’t think that their results actually support their main conclusion (suggestion) that the long-distance enhancers may contribute to allelic expression of Peg3 and Zim1.

Major comments

1. Imprinted domains are very unique because expression profiles of both of the paternal and maternal alleles are different. Therefore, the authors need to carry out 3C experiment on paternal and maternal alleles separately. The authors insisted that the observed ECR’s interaction with Peg3 and Zim1 are likely allele-specific: Peg3 on the paternal and Zim1 on the maternal allele. However, it is not guaranteed.

In the U1/KO2 mice with maternally expressed Peg3 and paternally expressed Zim1, the interaction of ECR’s and Peg3/Zim1 promoters (Fig. 6B) is far from the reversed version of wild type (Fig. 6A), suggesting that the pattern of interaction between the ECR’s and Peg3/Zim1 promoters is more complex and represents mixed pattern of both of the paternal and maternal interaction. Therefore, they should present the interaction profiles of paternal and maternal alleles in the U1/KO2 case, separately.

------- Response: We are very certain that the interaction of Peg3 with ECRs is paternal-specific and the interaction of Zim1 with ECRs is maternal-specific. In particular, the U1/KO2 sample has only one allele for the Peg3 promoter, which is the maternal allele in this case. Thus, the results from the Peg3 promoter represent the interaction of the maternal allele of Peg3 promoter with ECRs. If the interaction between Peg3 or Zim1 and ECRs is purely based on the promoter strength and associated properties of each gene, but not influenced by ECRs, we should have seen the reversed patterns of 3C results between WT and U1/KO2. However, the results turned out to be far from the reversed patterns, which were also noted by the reviewer. In fact, this has been one of the main evidence supporting our conclusion that ECRs are not biallelic but rather allele-biased. 

We also confirmed the allele-specific interaction between Peg3 promoter and ECRs by performing 3C with the samples from the mutant with the paternal deletion of the Peg3 promoter so that the base primer for Peg3 (Peg3p) binds to only one allele, which is the maternal allele. This control experiment did not show any enrichment or interaction, indicating that the repressed maternal allele of Peg3 is not normally available for the interaction with ECRs. This is also the case for Zim1 promoter: the repressed paternal allele of Zim1 promoter is not available for the interaction with ECRs. This set of control experiments has been included as Supporting information 9.

2. The authors suggested that the long-distance enhancers may contribute to allelic expression of Peg3 and Zim1, however, which ones contribute to the change of Peg3/Zim1 allelic expression although I agree that the expression levels of Peg3 and Zim1 are affected by some of the ECR’s by modifying chromosome conformation around Peg3 and Zim1 promoters?

------- Response: We believe that the ones showing allele-biased properties in 3C and DNA methylation should be responsible for the observed differences. So, ECR7, ECR11, ECR17, ECR19 should be the potential candidates.

3. DNA methylation analysis was carried out only using wild type mice but not using U1/KO2 mice. However, it is very important to know the DNA methylation status of the ECR2 and ECR17 in U1/KO2 mice because their methylation difference may cause change of interaction between these ECRs and Peg3/Zim1 promoters.

------- Response: We have tested the majority of ECRs in U1/KO2 and also in U1 and KO2 animals, and the results indicated no major difference in DNA methylation levels of the ECRs in various mutant alleles. Thus, we believe that the DNA methylation levels of ECRs are independent of these mutations.

4. I don’t think that interaction between ECRs and Peg3/Zim1 promoter in Fig. 8 reflects the authors experimental data and their explanation in the text. For one example, why do the fourth type (ECR3, ECR10 and ECR16) tended to show higher enrichment levels on the switched alleles with both promoter (the maternal allele for Peg3 and the paternal allele for Zim1) is classified by biallelic interaction because no enrichment was observed in the wild type (Fig. 6).

------- Response: I agree with the reviewer on this point. We have modified the figure accordingly.

Minor comments

1. Although it is necessary to explain the generation of a mouse mutant (U1/KO2), Figure 1 may be redundant because it’s the same as the previous report (ref. 18). The authors should include the information of the position of U1 promoter in the ECR region.

------- Response: Although redundant, we feel that the current manuscript should stand by itself as a separate unit for the readers, so we have included this particular figure. We have included the position information of U1 relative to the positions of ECRs in the legend of Fig 5.

2. In Fig. 4 (on DNA methylation levels of ECRs), authors described that ECR2 and ECR17 exhibited different levels of DNA methylation between paternal and maternal alleles. However, it is difficult to say that the difference of methylation levels are significant between two alleles because it was provided the data of one sample.

------- Response: We agree with the reviewer on this point. However, given our previous results, the DNA methylation methylation levels seem to be quite stable among biological replicates, since this analysis usually captures the average methylation levels of million cells. Thus, we are very confident that the results presented in this study is likely reproducible.

3. In Figure 3, images of immunostaining are low resolution.

------- Response: The images with higher resolution have been included as Supporting information 4.

Reviewer #3: Allele-specific enhancer interaction at the Peg3 imprinted domain

The authors look at the role of ECRs in the control of imprinted gene expression in the Peg3 domain.

They also investigate the consequences of a double switch so the paternal chromosome behaves maternally

and the maternal chromosome behaves paternally.

Major comments.

For the gene expression data shown in Figure 1 only 8 animals appear to have been studied, this is far too few to draw conclusions from and perform meaningful statistical analyses. The error bars shown are for technical replicates - I expect on biological replicates they would be much greater.

------- Response: Yes, the error bars are for technical replicates, but we have repeated each set of 3C experiments with 2 to 3 sets of biological replicates, which have been included as Supporting information 5-7. We are reporting only the results that are consistent and reproducible among these biological replicates. 

Are the wildtype samples littermates of the mutants? The expression levels of mutants should be normalised to wildtype littermates to ensure they had the same in utero environment and they are exactly the same age.

------- Response: Yes, the WT animals were all littermates of the U1/KO2 animals.

Also, it is not clear if the genetic background of the chromosome that each deletion is on is the same. Many knockouts are derived on non-C57B/6 strains and the results seen could purely be down to genetic background differences.

------- Response: The U1 allele has been derived from the C57BL/6J background, whereas the KO2 allele from 129/SvJ. The animals with the KO2 allele have been backcrossed more than 10 generations with C57BL/6J. Now, both alleles are on the same genetic background, thus we do not anticipate any artifacts that might be caused by genetic differences.

For the methylation analyses the authors used C57BL/6J females crossed with PWD/PhJ males. However,

for robust methylation analysis reciprocal crosses should be used to eliminate any genetic background effects.

------- Response: Yes, we agree with the reviewer on this point. At the same time, we do not believe that the allelic methylation level differences observed from the two ECRs were caused by genetic background differences based on all the previous results that have been derived from the same cross (Kim et al, 2012; He et al, 2016 and 2017)

In the 3C analysis the authors do not explain how they are able to distinguish the maternal and paternal alleles. Is this through SNPs or are the base primers located within the deletion so they are only interrogating interactions on one chromosome? As this provides a major part of their discussion their methods should be described more fully.

------- Response: The base primer for Peg3 is localized within the deleted region of the KO2 allele, thus the interaction of Peg3 in U1/KO2 was derived from the maternal allele. In the case of WT animals, we are very confident that the interactions of the Peg3 promoter with ECRs were derived from the paternal allele based on the following control experiment. We performed a set of control experiments with the animals with the paternal deletion of KO2, lacking the paternal allele of Peg3 promoter. In this series of control experiments, the interaction of Peg3 promoter with ECRs on the maternal allele was very minimal or marginal, indicating that the repressed maternal allele of Peg3 promoter is not available for the interaction with ECRs. Thus, the interactions of Peg3 with ECRs in the WT animals were mostly derived from the paternal allele. This is also the case for the paternal allele of Zim1 promoter. Thus, we are confident that the 3C experiments with two base primers measure allele-specific interaction with ECRs. This has been included as Supporting information 9.

In Figure 5, the middle panel for the Peg3 promoter wild-type - the band for ECR17 is much larger than expected - why is this?

------- Response: We confirmed later that this band represents the genuine interaction between Peg3 and ECR17, but not some artifactual fragment amplified from an unrelated locus. The different-size product was found to have been amplified due to another NcoI site nearby ECR17.

They mention a further set of 3Cs in KO2 pups lacking the paternal Peg3 allele - these data should be in the supplementary material.

------- Response: We are providing this result as Supporting information 9.

The authors mention that the 3C analyses were repeated twice. The data from all three biological replicates should be combined for the main figures rather that showing them separately. This would allow the statistical analyses to be performed on biological replicates rather than technical replicates (as I assume are being shown in Figure 6).

------- Response: It was not feasible to perform statistical analyses on the combined results of biological replicates, mainly due to the fact that the quality of 3C libraries varied quite a bit among individual biological replicates. 

Minor comments

What do the authors mean by "placental mammals". To make it clearer authors should use the accepted

terminology:

Eutherians - mammals that are not marsupials or monotremes

Therians - eutherian and marsupials

------- Response: Yes, we have replaced placental mammals with eutherians.

Line 5 should read eutherian evolution - as only 5 genes are known to be imprinted in marsupials

------- Response: Yes, we have corrected this.

Line 9 should read the majority of imprinted genes are found only in eutherian mammals.

------- Response: Yes, we have corrected this.

Line 10 should read imprinting may have evolved (or emerged) in the therian lineage. - A biological process can not be invented>

------- Response: Yes, we have corrected this.

Line 11 the sentence beginning "The mechanisms through imprinting " needs re-wording as it does not make sense.

------- Response: We have removed this sentence.

In paragraph 3 of the introduction the authors should say that the ECRs are located within the Usp29 transcript.

------- Response: Yes, we have included this information.

The orientation of the region in figures switches from figure to figure which make it difficult for a non-expert in this region to follow what is going on.

------- Response: We have modified Fig 1 to display the consistent orientation of the Peg3 domain as shown in the other figures.

---

## [Decision Letter · Decision Letter 1]

2 Oct 2019

PONE-D-19-17945R1

Allele-specific enhancer interaction at the Peg3 imprinted domain

PLOS ONE

Dear Dr. Kim,

Thank you for submitting your manuscript to PLOS ONE. After careful consideration, we feel that it has merit but does not fully meet PLOS ONE’s publication criteria as it currently stands. Therefore, we invite you to submit a revised version of the manuscript that addresses the points raised during the review process.

Your revised manuscript was reviewed by two of the original reviewers, who appreciated the corrections you had made, but thought that there were still some points that needed addressing.  In particular, reviewer 3 considers that the numbers of animals used does not justify the conclusions made, so could you please specifically address that point, or perhaps tone down your conclusions?

We would appreciate receiving your revised manuscript by Nov 16 2019 11:59PM. To enhance the reproducibility of your results, we recommend that if applicable you deposit your laboratory protocols in protocols.io, where a protocol can be assigned its own identifier (DOI) such that it can be cited independently in the future. For instructions see: http://journals.plos.org/plosone/s/submission-guidelines#loc-laboratory-protocols

We look forward to receiving your revised manuscript.

Kind regards,

Keith William Brown, Ph.D.

Academic Editor

PLOS ONE

Reviewers' comments:

Reviewer's Responses to Questions

**Comments to the Author**

1. If the authors have adequately addressed your comments raised in a previous round of review and you feel that this manuscript is now acceptable for publication, you may indicate that here to bypass the “Comments to the Author” section, enter your conflict of interest statement in the “Confidential to Editor” section, and submit your "Accept" recommendation.

Reviewer #1: (No Response)

Reviewer #3: (No Response)

2. Is the manuscript technically sound, and do the data support the conclusions?

Reviewer #1: Yes

Reviewer #3: Partly

3. Has the statistical analysis been performed appropriately and rigorously? 

Reviewer #1: Yes

Reviewer #3: No

4. Have the authors made all data underlying the findings in their manuscript fully available?

Reviewer #1: Yes

Reviewer #3: Yes

5. Is the manuscript presented in an intelligible fashion and written in standard English?

Reviewer #1: Yes

Reviewer #3: Yes

6. Review Comments to the Author

Reviewer #1: The authors have made efforts to address most of my earlier comments. The few remaning points are as follows:

1. Introduction (p3, para 1). Is it true to say “The majority of imprinted genes are found only in placental mammals”?

-----Response: Yes, that is correct.

Reviewer reply: I think this needs clarification still. Are you indicating that imprinting is restricted to mammals among animals (which is not contentious) or do you mean to say that most imprinted genes do not have orthologues outside of mammals (which I would contend)?

2. Introduction (p3, para 2). In what context is the Peg3 domain “well conserved”, do you mean specifically between mouse and human?

------- Response: The Peg3 domain is well conserved among several mammals, including human, mouse, sheep, dog, and cow.

Reviewer response: Please make this clear, e.g. “The Peg3 imprinted domain is localized in 500-kb genomic interval in human chromosome 19q13.4/proximal mouse chromosome 7 that is evolutionarily well conserved among mammals”

3. Results (p7, para 2). You state that Peg3 expression is ubiquitous in adult brain then later that Zim1 is uniquely expressed in ependymal cells; one of these statements must be wrong.

------- Response: Peg3 is expressed in the majority of neuron cells in the hypothalamus, whereas the expression of Zim1 was more limited to the small areas, such as PVN and SON. Also, we detected high expression levels of Zim1 in the ependymal cells from the current study. Overall, we do not feel these statements are contracting to each other.

Reviewer response: Then this is still unclear.

4. Results (p8, para 1). You note that Zim1 expression is “very similar” to Peg3 in PVN and SON. If Peg3 expression is ubiquitous and Zim1 expression more restricted, ZIm1 expression must form a sub-set of Peg3 expression and I don’t see how this forms good evidence they “share long-distance enhancers”. In addition, you specifically indicate enhancers “localized in the middle 200-kb region” of the domain but this cannot be inferred from expression patterns alone.

------- Response: We agree with the reviewer on this point that we need to have additional evidence to support the statements. However, we also strongly believe that the following scenario is most likely. Although Peg3 expression is more ubiquitous than Zim1 expression in the hypothalamus, both genes share a similar spatial expression pattern, the expression within the PVN and SON areas. This unique spatial expression pattern between these two genes is a strong indication that they share a set of cis-regulatory elements, or enhancers. Further, according to the surveys of the ENCODE dataset and also our own results (Kim and Ye 2016), the middle 200-kb region has the majority of potential enhancers within this imprinted domain. Thus, we believe that the shared enhancers are likely localized within the middle 200-kb region.

Reviewer response: Your ‘beliefs’ need to be supported by briefly setting out the logic you have presented here.

Reviewer #3: The authors do not address the point raised about the expression data. In the response they mention 3C experiments rather than expression.

"but we have repeated each set of 3C experiments with 2 to 3 sets of biological replicates, which have been

included as Supporting information 5-7. We are reporting only the results that are consistent and reproducible among these biological replicates."

The point being raised was that the expression data from 4 KOs and 4 WTs is not enough and they have not addressed this. Have they repeat the expression data on more individuals? At least 6 animals from each genotype across a number of litters are required. The error bars should then reflect the differences in expression across multiple individuals NOT technical replicates. These data should all be incorporated into 1 graph and shown in Figure 1.

I previously asked what was the genetic background the mutations were generated in . This question is not about the background of the genome as a whole but rather the region in close proximity to the deletions. The authors should note that even after backcrossing for 10 generations the DNA in close proximity to the deletion (which is selected for) is unlikely to have had a crossover event and thus will still be 129/SvJ in the case of the KO2 mice. As the KO2 region is close to genes it is highly likely that the genes located on the KO2 chromosome are 129/SvJ. Therefore the U1/KO2 mice Zim1 will be on a SvJ background but in Wildtypes on BL6. Differences in the sequences of genes and cis regulatory elements could explain the subtle differences in expression levels in Zim1.

On page 9

"The observed ECR's interaction with Peg3 and Zim1 are also likely allele-specific: Peg3 on the paternal and Zim1 on the maternal allele"

This statement is naive: enhancers have been shown to interact with promoters even in tissues where the gene is not expressed (data from Eileen Furlong's lab). The authors show in Supp Fig.9 that upon paternal transmission of KO2 there is reduced interaction of Peg3 promoter with ECRs. But the Zim1 data in this cross is biallelic. To fully characterise the interactions the authors should also perform 3Cseq on maternal KO2 heterozygotes and to interrogate the paternal chromosome separately.

All the data presented in Figure 7 relies on the assumption of allele specific interactions in the WT and U1/KO2 mice but this has not been established in the 3Cs presented. Supp_Figure 9A clearly shows that some interaction occurs between the maternal Peg3 promoter and the ECRs on a gel even though it was not detected in the qPCRs.

In the light of this Figure 7 should be re-labelled to show the mouse genotype, wildtype or U1/KO2, NOT Mat or Pat ECRs to make it clear that these were not allele specific 3Cs.

All the minor comments have been addressed except on page 3 they should say imprinting emerged in the therian lineage as imprinting has been observed in marsupials too.

7. PLOS authors have the option to publish the peer review history of their article (what does this mean?). If published, this will include your full peer review and any attached files.

Reviewer #1: No

Reviewer #3: No

---

## [Author Response · Author response to Decision Letter 1]

6 Oct 2019

Responses to Reviews' Comments

Reviewer #1: The authors have made efforts to address most of my earlier comments. The few remaining points are as follows:

1. Introduction (p3, para 1). Is it true to say “The majority of imprinted genes are found only in placental mammals”?

Response: Yes, that is correct.

Reviewer reply: I think this needs clarification still. Are you indicating that imprinting is restricted to mammals among animals (which is not contentious) or do you mean to say that most imprinted genes do not have orthologues outside of mammals (which I would contend)?

-----Response: We mean to say that the majority of imprinted genes do not have orthologues outside of mammals.

2. Introduction (p3, para 2). In what context is the Peg3 domain “well conserved”, do you mean specifically between mouse and human?

Response: The Peg3 domain is well conserved among several mammals, including human, mouse, sheep, dog, and cow.

Reviewer response: Please make this clear, e.g. “The Peg3 imprinted domain is localized in 500-kb genomic interval in human chromosome 19q13.4/proximal mouse chromosome 7 that is evolutionarily well conserved among mammals”

-----Response: We mean to say that the Peg3 domain is well conserved among several mammals, including human, mouse, sheep, dog, and cow. We have modified this sentence to further clarify this point.

3. Results (p7, para 2). You state that Peg3 expression is ubiquitous in adult brain then later that Zim1 is uniquely expressed in ependymal cells; one of these statements must be wrong.

Response: Peg3 is expressed in the majority of neuron cells in the hypothalamus, whereas the expression of Zim1 was more limited to the small areas, such as PVN and SON. Also, we detected high expression levels of Zim1 in the ependymal cells from the current study. Overall, we do not feel these statements are contracting to each other.

Reviewer response: Then this is still unclear.

-----Response: Our understanding is that two genes, Peg3 and Zim1, share several enhancers, but each of these two genes can have its own unique expression pattern in specific cell types, as shown in the ependymal cells of the hypothalamus.

4. Results (p8, para 1). You note that Zim1 expression is “very similar” to Peg3 in PVN and SON. If Peg3 expression is ubiquitous and Zim1 expression more restricted, ZIm1 expression must form a sub-set of Peg3 expression and I don’t see how this forms good evidence they “share long-distance enhancers”. In addition, you specifically indicate enhancers “localized in the middle 200-kb region” of the domain but this cannot be inferred from expression patterns alone.

Response: We agree with the reviewer on this point that we need to have additional evidence to support the statements. However, we also strongly believe that the following scenario is most likely. Although Peg3 expression is more ubiquitous than Zim1 expression in the hypothalamus, both genes share a similar spatial expression pattern, the expression within the PVN and SON areas. This unique spatial expression pattern between these two genes is a strong indication that they share a set of cis-regulatory elements, or enhancers. Further, according to the surveys of the ENCODE dataset and also our own results (Kim and Ye 2016), the middle 200-kb region has the majority of potential enhancers within this imprinted domain. Thus, we believe that the shared enhancers are likely localized within the middle 200-kb region.

Reviewer response: Your ‘beliefs’ need to be supported by briefly setting out the logic you have presented here.

-----Response: We have modified this paragraph to make more logical as suggested. 

Reviewer #3: The authors do not address the point raised about the expression data. In the response they mention 3C experiments rather than expression.

"but we have repeated each set of 3C experiments with 2 to 3 sets of biological replicates, which have been included as Supporting information 5-7. We are reporting only the results that are consistent and reproducible among these biological replicates."

The point being raised was that the expression data from 4 KOs and 4 WTs is not enough and they have not addressed this. Have they repeat the expression data on more individuals? At least 6 animals from each genotype across a number of litters are required. The error bars should then reflect the differences in expression across multiple individuals NOT technical replicates. These data should all be incorporated into 1 graph and shown in Figure 1.

-----Response: We have used these 4 sets of KO and WT derived from the previous breeding experiments, which took place 2 years ago throughout a span of 6 months. This was the maximum number of WT and KO littermate pairs with matching age at that time. Although we understand the reviewer's point, it is not feasible to perform another round of breeding experiments, which may take another 6 month to 1 year. At the same time, we are very confident with the conclusion derived from this set of the results given the associated statistical significances. 

I previously asked what was the genetic background the mutations were generated in . This question is not about the background of the genome as a whole but rather the region in close proximity to the deletions. The authors should note that even after backcrossing for 10 generations the DNA in close proximity to the deletion (which is selected for) is unlikely to have had a crossover event and thus will still be 129/SvJ in the case of the KO2 mice. As the KO2 region is close to genes it is highly likely that the genes located on the KO2 chromosome are 129/SvJ. Therefore the U1/KO2 mice Zim1 will be on a SvJ background but in Wildtypes on BL6. Differences in the sequences of genes and cis regulatory elements could explain the subtle differences in expression levels in Zim1.

-----Response: When we first started working on this domain 20 years ago, we have assembled and sequenced two BAC contigs from the 129 and B6 strains. Some of these have been published (Kim et al, 2000 Genome Research 10:1138-1147; Kim et al, 2001 Genomics 74:129-141). According to this survey, we have not found any SNPs between these two strains in the ECRs, promoter and exon regions of the imprinted genes, including Peg3/Usp29 and Zim1. Thus, we believe that the expression difference reported in the current manuscript cannot be caused by the sequence variations between B6 and 129.

On page 9

"The observed ECR's interaction with Peg3 and Zim1 are also likely allele-specific: Peg3 on the paternal and Zim1 on the maternal allele"

This statement is naive: enhancers have been shown to interact with promoters even in tissues where the gene is not expressed (data from Eileen Furlong's lab). The authors show in Supp Fig.9 that upon paternal transmission of KO2 there is reduced interaction of Peg3 promoter with ECRs. But the Zim1 data in this cross is biallelic. To fully characterise the interactions the authors should also perform 3Cseq on maternal KO2 heterozygotes and to interrogate the paternal chromosome separately.

All the data presented in Figure 7 relies on the assumption of allele specific interactions in the WT and U1/KO2 mice but this has not been established in the 3Cs presented. Supp_Figure 9A clearly shows that some interaction occurs between the maternal Peg3 promoter and the ECRs on a gel even though it was not detected in the qPCRs.

In the light of this Figure 7 should be re-labelled to show the mouse genotype, wildtype or U1/KO2, NOT Mat or Pat ECRs to make it clear that these were not allele specific 3Cs.

-----Response: We actually performed a series of 3C using the sample with the maternal transmission of KO2. However, the results cannot be used as a control demonstrating the allele-specific interaction mainly due to the fact that the maternal deletion of KO2 has a very unusual trans-allelic effects, up-regulating the remaining paternal allele (Bretz and Kim, 2018 PLoS ONE 13:e0206112).

Regarding re-labeling Fig 7, both WT and U1/KO2 have their own parental alleles (MAT and PAT). Thus, it will make very busy figures with many labelings. Thus, we included the following sentences in the figure legend. The values for the paternal and maternal alleles of Peg3 were derived from WT and U1/KO2, respectively. On the other hand, the values for the maternal and paternal alleles of Zim1 were derived from WT and U1/KO2, respectively. 

All the minor comments have been addressed except on page 3 they should say imprinting emerged in the therian lineage as imprinting has been observed in marsupials too.

---

## [Editor Report · Decision Letter 2]

10 Oct 2019

Allele-specific enhancer interaction at the Peg3 imprinted domain

PONE-D-19-17945R2

Dear Dr. Kim,

We are pleased to inform you that your manuscript has been judged scientifically suitable for publication and will be formally accepted for publication once it complies with all outstanding technical requirements.

With kind regards,

Keith William Brown, Ph.D.

Academic Editor

PLOS ONE
---

## [Editor Report · Acceptance letter]

14 Oct 2019

PONE-D-19-17945R2 

Allele-specific enhancer interaction at the *Peg3* imprinted domain 

Dear Dr. Kim:

I am pleased to inform you that your manuscript has been deemed suitable for publication in PLOS ONE. Congratulations! Your manuscript is now with our production department. 

With kind regards,

on behalf of

Dr. Keith William Brown 

Academic Editor

PLOS ONE